# Steerable Equivariant Representation Learning

## Abstract

Pre-trained deep image representations are useful for post-training tasks such as classification through transfer learning, image retrieval, and object detection. Data augmentations are a crucial aspect of pre-training robust representations in both supervised and self-supervised settings. Data augmentations explicitly or implicitly promote *invariance* in the embedding space to the input image transformations. This invariance reduces generalization to those downstream tasks which rely on sensitivity to these particular data augmentations. In this paper, we propose a method of learning representations that are instead *equivariant* to data augmentations. We achieve this equivariance through the use of *steerable* representations. Our representations can be manipulated directly in embedding space via learned linear maps. We demonstrate that our resulting steerable and equivariant representations lead to better performance on transfer learning and robustness: e.g. we improve linear probe top-1 accuracy by between 1% to 3% for transfer; and ImageNet-C accuracy by upto 3.4%. We further show that the steerability of our representations provides significant speedup (nearly 50×) for test-time augmentations; by applying a large number of augmentations for out-of-distribution detection, we significantly improve OOD AUC on the ImageNet-C dataset over an invariant representation.

## 1 Introduction

Embeddings of pre-trained deep image models are extremely useful in a variety of downstream tasks such as zero-shot retrieval (Radford et al., 2021), few-shot transfer learning (Tian et al., 2020), perceptual quality metrics (Zhang et al., 2018) and the evaluation of generative models (Heusel et al., 2017; Salimans et al., 2016). The pre-training is done with various supervised or self-supervised losses (Khosla et al., 2020; Radford et al., 2021; Chen et al., 2020) and a variety of architectures (He et al., 2016; Dosovitskiy et al., 2020; Tolstikhin et al., 2021). The properties of pre-trained embeddings, such as generalization (Zhai et al., 2019) and robustness (Naseer et al., 2021), are therefore of significant interest. Most current pre-training methods impose invariance to input data augmentations either via losses (Tsuzuku et al., 2018; Chen et al., 2020; Caron et al., 2021) or architectural components such as pooling (Fan et al., 2011). For invariant embeddings, the (output) embedding stays nearly constant for all transformations of a sample (e.g. geometric or photometric transformations of the input). Invariance is desirable for tasks where the transformation is a nuisance variable (Lyle et al., 2020). However, prior work shows that it can lead to poor performance on tasks where sensitivity to transformations is desirable (Dangovski et al., 2022; Xiao et al., 2021).

Formally, let $e(\mathbf{x}; \mathbf{w})$ represent the encoder network that maps an input sample $\mathbf{x}$ (image) to the embedding $e$, where $\mathbf{w}$ are the parameters of the network. We use $e(\mathbf{x})$ and $e(\mathbf{x}; \mathbf{w})$ interchangeably for ease of notation. The data augmentation of a sample $\mathbf{x}$ is represented as $g(\mathbf{x}; \theta)$, often shortened to $g(\mathbf{x})$ for brevity. $\theta$ refers to the parameters of the augmentation, e.g. for photometric transformations it is a 3-dimensional vector of red, green and blue shifts applied to the image. Given this notation, if $e(g(\mathbf{x}; \theta)) = e(\mathbf{x})$, i.e. the embedding does not change due to the input transformation $g(\theta)$, it is said to be invariant to $g$.

Equivariance is a more general property: intuitively it means that applying a transformation to the input changes the embedding in a predictable, well-defined manner. Consider $M$ a map of the embedding space onto itself such that $e(g(\mathbf{x}; \theta)) = M(e(\mathbf{x}), \theta)$, then the embedding is said to be equivariant to augmentation $g$ with tranformation $M$. In other words, the actions of $g$ in input space commutes with the action of $M$ in

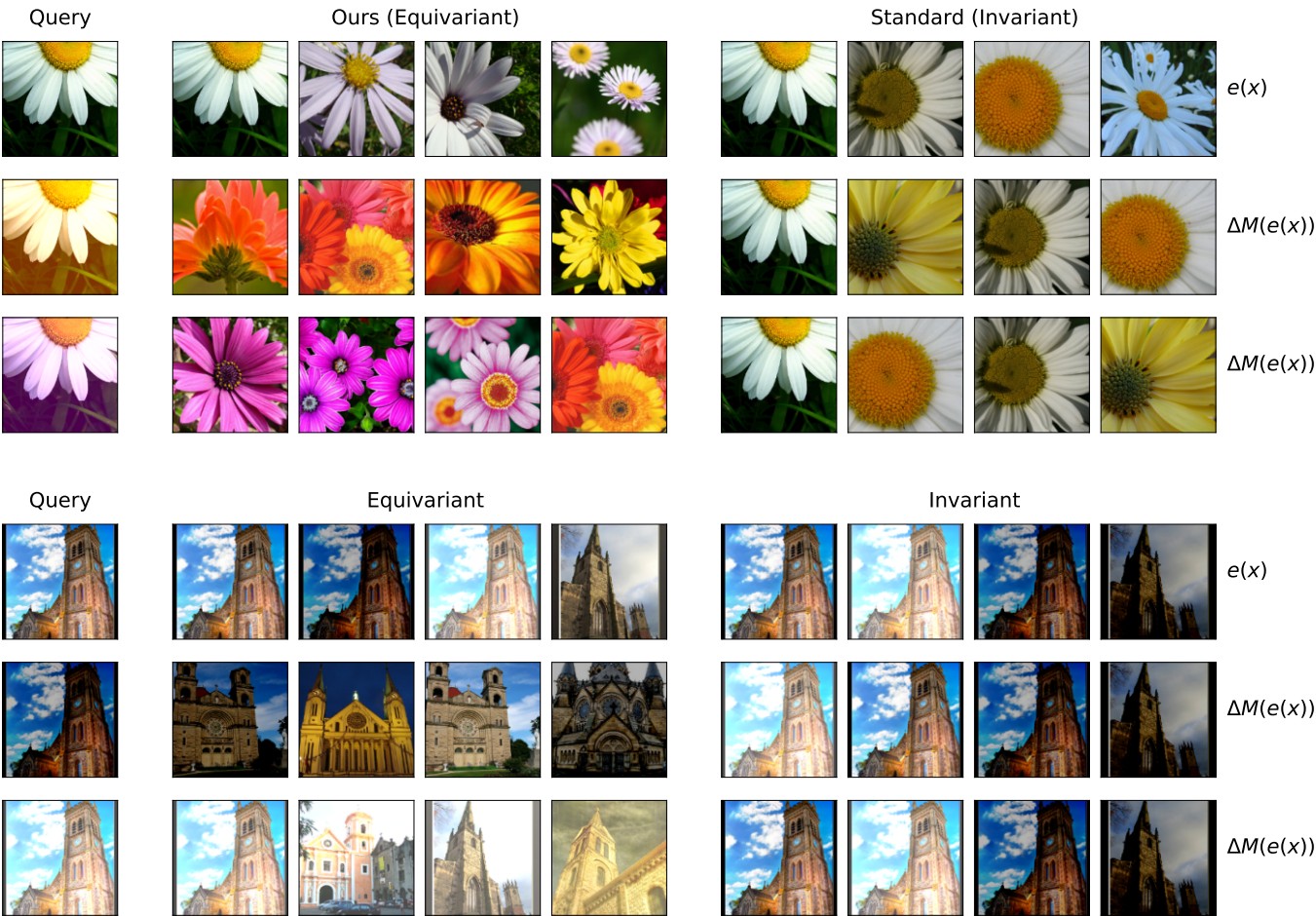

Figure 1: Two examples of image retrieval, comparing our steerable equivariant model to the baseline invariant (standard) model. The top example (flowers) is for color-based steering; the bottom example (buildings) is shown for brightness-based retrieval. For each example, we show three query images in the left column, along with nearest neighbors in the next 8 columns (4 each for the steerable and standard models). Please see text for definitions of $e(\mathbf{x})$, $M(e(\mathbf{x}))$ and $\Delta M(e(\mathbf{x}))$. Query image shown is simply for illustration; we do *not* use that image for the retrieval. The steerable model retrieves images where the color or brightness change overrides semantics. For example, the second query on the flowers example retrieves yellow/pink neighbors and the third query retrieves purple/blue colored flowers; similarly for a darker image in the second example, darker images are retrieved and for brighter examples, brighter images are retrieved. The invariant model retrievals are fairly static between different color or brightness changes.

embedding space. If $M$ were the identity function, then we recover invariance. If the map $M$ is well-behaved, imposing equivariance throughout training encourages the latent space embedding to change smoothly with respect to $\theta$, the parameters of the transformation, on application of $g$ to the input. $M$ can be parameterized by a linear (a matrix) or a nonlinear function (deep network), the output of which is another vector of the same dimensions as $e(\mathbf{x})$. We use simple linear functions for ease of optimization and computational efficiency, as is highlighted in Section 4.4.

Explicit acces to $M$ (in contrast to a garantee of existence) allows to directly manipulate the embeddings $e$, leading us to the concept of *steerability* (e.g.(Freeman et al., 1991)): "it is possible to determine the response of a filter of arbitrary orientation without explicitly applying that filter". In our case, steerability thus refers to the ability to predict the effect of the transformation $g$ on the embedding directly, via the application of $M$ in embedding space, without having to explicity use the encoder network on $g(x)$. In other words, we are able to "*steer*" our embeddings to apply input-equivalent transformations directly in embedding space.

It has been shown that pre-trained embeddings often accommodate linear vector operations to enable e.g. nearest neighbor retrieval using Euclidean distance (Radford et al., 2021); this is a coarse form of steerability. However, without more explicit control on the embeddings, it is difficult to perform fine-grained operations on this vector space, for example, re-ordering retrieved results by color attributes. It is not very useful in practice to steer an invariant model: the embeddings may change very little in response to steering. However, enabling steerability for an equivariant embedding opens up a number of applications for control in embedding space; we show the benefits in our experiments.

We demonstrate the concepts of invariance, equivariance, and steerability of embeddings with a visual depiction in Figure 2. Intuitively, equivariance represents the property of an embedding to transform predictably in response to a transformation of the input. When this response is trivially the identity function, an embedding is invariant to the transformation, otherwise it is equivariant. Steerability refers to the ability to predict and apply this response directly in the space of the embeddings. It is possible to have an equivariant representation for which the form of this response function is unknown, and hence it is not possible to have steerability.

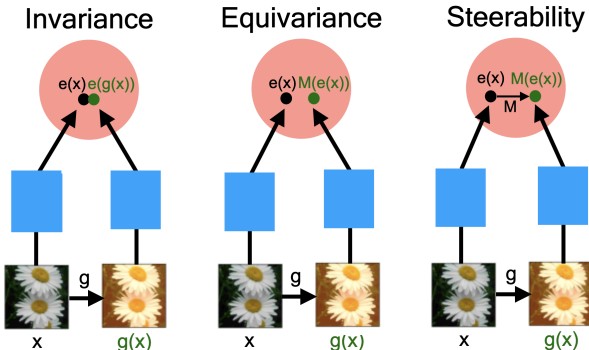

Figure 2: The concepts of invariance, equivariance and steerability of embeddings. Blue boxes represent the (shared) encoder that takes the input $\mathbf{x}$ to the embedding $e(\mathbf{x})$. $g(\mathbf{x})$ represents a transformation of $\mathbf{x}$ in input space; and $M(e(\mathbf{x}))$ is a mapping in embedding space. Equivariance is a necessary but not sufficient condition for steerability.

In prior work, equivariant embeddings have been shown to have numerous benefits: reduced sample complexity for training, improved generalization and transfer learning performance (Cohen & Welling, 2016b; Simeonov et al., 2021; Lenssen et al., 2018; Xiao et al., 2021). Equivariance has usually been achieved by the use of architectural modifications (Finzi et al., 2021; Cohen & Welling, 2016b), which are mostly restricted to symmetries represented as matrix groups. However, this does not cover important transformations such as photometric changes or others that cannot be represented explicitly as matrix transformations. Xiao et al. (2021) and Dangovski et al. (2022) propose more flexible approaches to allow for using more general input transformations. However, a key distinction between these works and ours is that we parameterize the transformations $M$ in latent space and learn them, allowing for *steering*. The other methods enforce equivariance without explicitly defining the latent space transformations, and are thus not immediately usable for steerability. Furthermore, our method is generalizable across architectures and pre-training losses, whereas the others are restricted to self-supervised contrastive loss paradigms.

We introduce a simple and general regularizer to encourage embedding equivariance to input data augmentations. The same mechanism (mapping functions) used for the regularization enables a simple steerable mechanism to control embeddings post-training. Our regularizer achieves significantly more equivariance (as measured by the metric in (Jayaraman & Grauman, 2015)) than pre-training without the regularizer. Prior work (Cohen & Welling, 2016a; Deng et al., 2021; Zhang, 2019) has introduced specialized architectures to make deep networks equivariant to specific transformations such as rotation or translation. Our approach is complementary to these works: as long as a transformation is parameterized in input space, we can train a mapping function in embedding space to mimic that transformation. It is agnostic to architecture. We show the benefits of our approach with applications in nearest neighbor retrieval when different augmentations are applied *in embedding space*, showing the benefits of steerable control of embeddings (see Fig. 1 as an

example). We also test our approach for out-of-distribution detection, transfer, and robustness: our steerable equivariant embeddings significantly outperform the invariant model. To summarize, our contributions in this work are (a) introducing a new, more flexible method, to make representations equivariant to transformations, (b) introducing steerability to representations, (c) demonstrating novel benefits of equivariant representations over invariant, and (d) demonstrating that these benefits are enhanced and more efficient with steerability.

## 2 Related Work

Data augmentation is a crucial component of modern deep learning frameworks for vision. Pre-defined transformations are chained together to create new versions of every sample fed into the network. Examples of such transformations are random image crops, color jitter, mixing of images and rotations (Cubuk et al., 2020; Hendrycks et al., 2019; Yun et al., 2019; Hendrycks et al., 2019; Chen et al., 2020; Gidaris et al., 2018). Most data augmentation pipelines involve randomly picking the parameters of each transformation, and defining a deterministic (Cubuk et al., 2020) or learned order of the transformations (Cubuk et al., 2018). Adversarial training is another class of augmentations, (Xie et al., 2020; Herrmann et al., 2021) which provide model-adaptive regularization. Data augmentation expands the training distribution to reduce overfitting in heavily over-parametrized deep networks. This provides a strong regularization effect, and improves generalization consistently, across architectures, modalities and loss functions (Hernández-García & König, 2018; Steiner et al., 2021; Hou et al., 2018; Shen et al., 2020; Chen et al., 2020; Caron et al., 2020; He et al., 2021). Data augmentations are crucial in the training of self-supervised contrastive losses (Chen et al., 2020; Grill et al., 2020).

Most losses for deep networks implicitly or explicitly impose invariance to input data augmentations (Tsuzuku et al., 2018). For all transformations of a sample (e.g. different color variations), the output embedding stays nearly constant. When this property is useful (e.g. classification under perturbations), invariance is desirable (Lyle et al., 2020). A number of papers have studied invariance properties of convolutional networks to specific augmentations such as translation (Azulay & Weiss, 2018; Zhang, 2019; Bruna & Mallat, 2013), rotation (Sifre & Mallat, 2013) and scaling (Xu et al., 2014). These architectural constructs have been made somewhat redunrdant in newer Transformer-based deep networks (Vaswani et al., 2017; Dosovitskiy et al., 2020; Tolstikhin et al., 2021) which use a mix of patch tokens and multilayer perceptrons. However, invariance is not universally desirable. Equivariance is a more general property from which invariance can be extracted by using aggregation operators such as pooling (Laptev et al., 2016; Fan et al., 2011). Equivariant architectures have benefits such as reduced sample complexity in training (Esteves, 2020) and capturing symmetries in the data (Smidt, 2021). Rotational equivariance has been extensively studied for CNN's (Cohen & Welling, 2016a; Simeonov et al., 2021; Deng et al., 2021). Convolutional networks (without pooling) are constructed to have translational equivariance, although papers such as (Azulay & Weiss, 2018) try to understand when this property does not hold. A number of works have suggested specific architectures to enable equivariance e.g. (Cohen & Welling, 2016b; Dieleman et al., 2016; Lenssen et al., 2018; Finzi et al., 2020; Sosnovik et al., 2019; Romero et al., 2020; Romero & Cordonnier, 2020; Bevilacqua et al., 2021; Smets et al., 2020). However, these architectures have not been widely adopted in spite of their useful properties, possibly due to the extra effort required to setup and train such specialized models.

For many applications, it is also useful to be able to *steer* equivariant embeddings in a particular direction, to provide fine-grained control. While it is a well-understood concept in signal and image processing (Freeman et al., 1991), it is less widely applied in neural networks. Our work is inspired by that of (Jayaraman & Grauman, 2015), who introduce an unsupervised learning objective to tie together ego-motion and feature learning. The resulting embedding space captures equivariance of complex transformations in 3-D space, which are hard to encode in architectures directly: they use standard convolutional networks and an appropriate loss to encourage equivariance. They show significant benefits of their approach for downstream recognition tasks over invariant baselines. The works of (Xiao et al., 2021; Dangovski et al., 2022) are also closely related. They build explicit equivariant spaces for specific data augmentations (in their case, color jitter, rotation and texture randomization). However, they do so indirectly by increasing 'sensitivity' to transformations, and do not build any $M_g$ equivalent maps. They hence do not provide steerability. Additionally, their work is restricted to the contrastive setting, whereas our regularizer can be added to any training paradigm.

## 3 Model

In this paper, we work in the context of supervised training on ImageNet classification models. However, note that our approach is general and easily extends to self-supervised settings. e.g. (Chen et al., 2020; He et al., 2021; Chen et al., 2021). Our standard (invariant) model is trained with a cross-entropy loss, along with weight decay regularization with hyperparameter $\lambda$:

$$L_{CE}(\mathbf{x}) = \sum_c -\log p_c(\mathbf{x}; \mathbf{w}) y_c(\mathbf{x}) + \lambda \|\mathbf{w}\|_2^2 \tag{1}$$

Here, the embedding $e(\mathbf{x}; \mathbf{w})$ is projected to a normalized probability space $p(\mathbf{x}; \mathbf{w})$ (the "logits" layer of the network). $y(\mathbf{x})$ refers to the target label vector for the sample $\mathbf{x}$, used for supervised learning. Vector components in both are indexed by $c$, which can $y_c(\mathbf{x})$ refers to , say, the classes for supervised training the $c$'th entry of the vector $y(\mathbf{x})$. The entries of $p(\mathbf{x})$ and $y(\mathbf{x})$ are between 0 and 1, and they sum to 1 to form the parameters of a categorical distribution.

The usual manner of training cross-entropy loss is to first apply a sequence of data augmentations to $\mathbf{x}$ e.g. (Cubuk et al., 2020) and then pass the transformed version of $\mathbf{x}$ into the network. Since all transformations of $\mathbf{x}$ are encouraged to map to the same distribution $y(\mathbf{x})$, this loss promotes *invariance* in $p(\mathbf{x})$ and therefore also in the embedding $e(\mathbf{x})$.

### 3.1 Measuring Equivariance for an Augmentation

In works such as Cohen & Welling (2016a); Lenssen et al. (2018), the architecture guarantees equivariance. Our approach is agnostic to architecture, so we desire a quantitative way to measure the equivariance with respect to a particular augmentation. We adapt the measure in (Jayaraman & Grauman, 2015). Denoting a given augmentation by $a$, we use the formula:

$$\rho_a(\mathbf{x}) = \frac{\|M_a(e(\mathbf{x}), \theta_a) - e(g_a(\mathbf{x}; \theta_a))\|_2}{\|e(g_a(\mathbf{x}, \theta_a)) - e(\mathbf{x})\|_2} \tag{2}$$

The denominator measures invariance: lower values mean more invariant embeddings $e$ w.r.t. the augmentation $a$ applied to input $\mathbf{x}$. The numerator measures equivariance: we want the embedding of a transformation $g_a(\mathbf{x})$ to be represented as a transformation in embedding space, represented by $M_a(e(\mathbf{x}))$. The lower the value of $\rho_a(\mathbf{x})$, the more equivariant the embedding $e(\mathbf{x})$. Note that we need a ratio in Eqn. 2, rather than just the numerator, to exclude trivial solutions where $g_a(\mathbf{x}; \theta_a)$ is mapped to the same point for all $\theta_a$ (for a given $\mathbf{x}$), and similarly for $M_a(e(\mathbf{x}))$. This would indeed make the numerator small, but it is representative of invariance rather than equivariance. Dividing by the denominator ensures that $e(g_a(\mathbf{x}; \theta_a))$ be distinct from $e(\mathbf{x})$ for different values of $\theta_a$. Note that $M_a$ and $g_a$ share the same transformation parameter $\theta_a$. We note that $\rho_a(\mathbf{x})$ is equivalent to the Expected Group Sample Equivariance metric in Gruver et al. (2022). The paper shows that this metric shows similar trends as the Lie Derivative, another metric of local equivariance. We thus expect $\rho_a(\mathbf{x})$ to show similar trends as the Lie Derivative as well.

### 3.2 Equivariance-promoting Regularizer

We use the numerator of Eqn. 2 to define a regularizer to promote equivariance in the embeddings:

$$L_E^a(\mathbf{x}) = \|M_a(e(\mathbf{x}), \theta_a) - e(g_a(\mathbf{x}; \theta_a))\|_2^2 \tag{3}$$

where we have a separate term for each augmentation $a$ that is applied at the input. As before, the transformation-specific parameters $\theta_a$ are shared between the embedding map $M_a$ and the input transformation $g_a$. Thus, the embedding map learns to apply a transformation in embedding space that mimics the effect of applying the transformation to the input. Hence, these maps $M$ allow us to directly manipulate the embeddings $e$ and provide fine-grained control w.r.t transformations: a concept we call *steerability*. Note that it is possible to make representations equivariant but not steerable, as in Xiao et al. (2021); Dangovski et al. (2022), by not recovering the $M_a$'s.

We use the following structure for $M_a$. A vector of continuous valued parameters $\theta_a$ is first mapped to a 128-dimensional intermediate parameter representation using a single dense layer without non-linearity. This vector is then concatenated with the given embedding $e(\mathbf{x})$ and passed through another dense linear layer, to give the final output of the map which is the same dimensionality (2048) as $e(\mathbf{x})$. This structure adds around 1% extra parameters to a ResNet-50 model.

### 3.3 Uniformity Regularizer

We observe that embeddings learned using $L_{CE}(\mathbf{x})$ lead to well-formed class clusters, but within a cluster they collapse onto each other, thereby increasing invariance (and Equation (3) cannot prevent this). To overcome this, we enforce a uniformity loss (Wang & Isola, 2020) which is given by:

$$L_U(\mathbf{x}) = \log \sum_{ij} \exp^{-\|e(\mathbf{x}_i) - e(\mathbf{x}_j)\|_2^2 / \tau} \tag{4}$$

$\tau$ is a temperature parameter often set to a small value such as 0.1. This loss encourages the embeddings of sample $\mathbf{x}_i$ to separate from embeddings of other samples $\mathbf{x}_j$. We find that it also has the effect of spreading out augmentations of the same sample, increasing the equivariance of the embeddings and reducing the likelihood of a trivial solution for the mapping $M_a$. This could alternatively have been achieved by maximizing the denominator of Equation (2), but we find empirically that strongly pushed $e(\mathbf{x})$ and $e(g_a(\mathbf{x}))$ apart and destabilized training.

### 3.4 Loss

Putting the above together, our final loss to train an equivariant/steerable model is given by:

$$L_{CEU}(\mathbf{x}) = L_{CE}(\mathbf{x}) + \alpha \sum_a L_E^a(\mathbf{x}) + \beta(L_U(\mathbf{x}) + \sum_a L_U(g_a(\mathbf{x}; \theta_a))) \tag{5}$$

The sums are computed over different augmentations $a$ for which we desire equivariance in the embedding, and a separate embedding map is learned for each. $\alpha$ and $\beta$ are weighting hyper-parameters. Note that the $L_E$ and $L_U$ terms are applied to the embedding $e$ and the cross-entropy is applied to the softmax output $p$.

## 4 Experiments

Our models are trained on the ImageNet dataset (Deng et al., 2009), on the ResNet-50 architecture (He et al., 2016). To train our steerable linear maps, we use the following augmentations:

- Geometric: a crop transformation parameterized by a 4-dimensional parameter vector $\theta_{geo}$, representing the position of the top left corner (x and y coordinates) of the crop, crop height and crop width. All values are normalized by the image size (224 in our case). When $\theta_{geo} = [0, 0, 1, 1]$, this corresponds to no augmentation. We denote the corresponding steerable map as $M_{geo}$. This augmentation encompasses random crop, zoom, resize (Chen et al., 2020), and translation (by only varying the top left corner).

- Photometric: a color jitter tranformation parameterized by a 3-dimensional parameter vector $\theta_{photo}$, which represents the respective *relative* change in the values of the RGB channels. The values of $\theta_{photo}$ are in the range of $[-1, 1]$. When $\theta_{photo} = [0, 0, 0]$, this corresponds to no augmentation. We denote the corresponding steerable map as $M_{photo}$. This augmentation encompasses global contrast, brightness, and hue/color transformations.

Both the standard (invariant) and our steerable equivariant models are trained with the same data augmentation. The invariant model is trained with the standard cross-entropy loss in Eqn. 1 for 250 epochs, with a batch size of 4096 (other training details are in the Appendix A.1). This model achieves a top-1 accuracy of 75.17% on the ImageNet eval set. The equivariant/steerable model is trained with the loss in Eqn. 5 with the

same learning rate schedule, number of epochs and batch size as the invariant model, with hyperparameters $\alpha$=0.1 and $\beta$=0.1. Ablations for selecting these hyperparameters are presented in Appendix Section A.2. As earlier, the data augmentations from SimCLR (Chen et al., 2020) are applied to a sample $\mathbf{x}$ and used in $L_{CEU}$. In addition to this, the parameters $\theta_{photo}$ and $\theta_{geo}$ are sampled uniformly at random within their pre-defined ranges. These are applied independently to generate two more of views of $\mathbf{x}$, for use in $L_E^a(\mathbf{x})$ and $\bar{L}_U(\mathbf{x})$. This model achieves a top-1 accuracy of 74.96% on the ImageNet evaluation set. To facilitate comparison with the equivariant model, we endow the invariant model with similar maps $M_{geo}$ and $M_{photo}$ for both augmentations. We train them using Eqn. 3 but with the encoder parameters frozen.

## 4.1 Equivariance Measurement

We use the trained maps (two maps for each model) to measure the equivariance of both the models using the measure of Eqn. 2, and report them in Table 1. We see that the equivariant/steerable model has significantly lower $\rho$ values than the baseline invariant model for both data augmentations, showing that pre-training with an equivariance promoting regularizer is crucial for learning equivariant and steerable representations.

| Model | $\rho_{geo} \downarrow$ | $\rho_{photo} \downarrow$ | Flowers-102 $\uparrow$ | DTD $\uparrow$ | Pets $\uparrow$ | Caltech-101 $\uparrow$ |
|---|---|---|---|---|---|---|
| Invariant (Standard) | 0.982 | 0.983 | 84.38 | 64.15 | 91.82 | 86.55 |
| Equivariant (Ours) | **0.474** | **0.658** | **87.17** | **65.15** | **92.13** | **87.23** |

Table 1: Left: Equivariance measure (Eqn. 2) for the two sets of augmentations. Right: Linear probe accuracy on 4 datasets: our equivariant model consistently outperforms the invariant model.

## 4.2 Nearest Neighbor Retrieval

A common use-case for pre-trained embeddings is their use in image retrieval (Xiao et al., 2021; Zheng et al., 2017). In the simplest case, given a query embedding $e_q$, the nearest neighbors of this embedding are retrieved using Euclidean distance in embedding space over a database of stored embeddings. We test qualitative retrieval performance of the invariant and equivariant models. To mimic a practical setting, we populate a database with the embeddings $e(\mathbf{x}_i)$ of samples $\mathbf{x}_i$ from the ImageNet validation set. All query and key embeddings are normalized to have an $l_2$ norm of 1, before being used in retrieval. Given a query sample $\mathbf{x}$, we consider the following query embeddings:

- $e(\mathbf{x})$: Embedding of the sample $\mathbf{x}$ with no augmentation applied.
- $e(g(\mathbf{x}))$: Embedding of $\mathbf{x}$ after we apply a transformation $g$ in input space.
- $M(e(\mathbf{x})))$: Embedding map applied to embedding $e(\mathbf{x})$ to steer towards $e(g(\mathbf{x}))$.
- $\Delta M(e(\mathbf{x}))$: Compute the difference between $e(\mathbf{x})$ and $M(e(\mathbf{x}))$ and add it back to $M(e(\mathbf{x}))$ with a weight $w_m$; i.e. $\Delta M(e(\mathbf{x})) = M(e(\mathbf{x})) + w_m(M(e(\mathbf{x})) - e(\mathbf{x}))$. This enables $\Delta M(e(\mathbf{x}))$ to be 'pushed' further in the direction of the transformation. $w_m$ is empirically chosen and set to 5 and 1 for the equivariant and invariant models respectively for all retrieval experiments.

In addition to qualitative results, we compute the mean reciprocal rank (MRR): $MRR = \frac{1}{n} \sum_{i=1}^{n} \frac{1}{r_i}$, where $r_i$ is the rank of the desired result within a list of retrieved responses to a single query, and $n$ is the number of queries. MRR lies in the range $[0, 1]$, with 1 signifying perfect retrieval. We calculate MRR for both models, for color, crop, and brightness augmentations. Table 2 shows that the equivariant model achieves better MRR across all augmentations.

We note that querying the nearest neighbours of an image can be thought of as a way to indirectly query the structure of the latent space. We also provide low-dimensional visualisations using t-SNE in Appendix A.3.

### 4.2.1 Photometric Augmentations

Results for color augmentation comparing the invariant and our equivariant/steerable models are displayed in Figure 3. We observe that retrieved results for both $e(g(\mathbf{x}))$ and $M(e(\mathbf{x}))$ change more in response to a

| Model | Color | Zoom | Bright. | Color-Crop (↑) |
|---|---|---|---|---|
| Invariant | 0.619 | 0.356 | 0.444 | 0.126 |
| Equivariant | **0.974** | **0.728** | **0.757** | **0.827** |

Table 2: Mean Reciprocal Rank for single (left 3) and composed (right) augmentations.

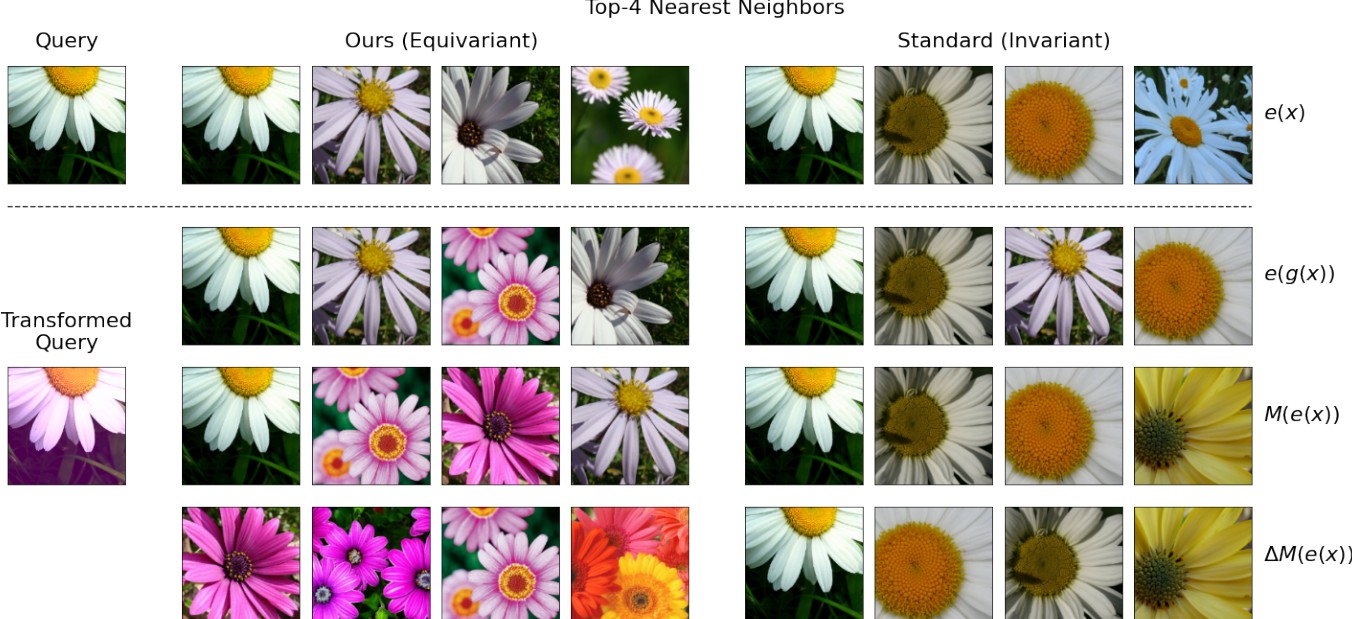

Figure 3: Nearest neighbour retrieval on original and transformed images. The leftmost column shows the query images, with the transformation applied in input space for the first two rows and the latent space for the next two. The remaining eight columns display top-4 nearest neighbours for each model. For both models, we show performance with 4 types of embeddings. $e(g(\mathbf{x}))$ and $M(e(\mathbf{x}))$ tend to be similar to each other (since Eqn. 3 encourages this). For the invariant model, semantic retrieval (flowers) override visual (pink color). The equivariant model can perform better visual retrieval. By steering using $\Delta M(e(\mathbf{x}))$, we can further enhance the color component of the embedding to control visual vs semantic retrieval.

change in query color for our steerable equivariant model than the invariant model. The color of the retrieved results for all queries for the standard model do not change appreciably, confirming invariance. This effect is even more pronounced for $\Delta M(e(\mathbf{x}))$. We were unable to find any value of the parameter $w_m$ for the invariant model that gave results qualitatively similar to the equivariant/steerable model. In Figures 1(top 3 rows) and 4, we show more examples across different classes and colors. Figure 1(bottom 3 rows) shows retrieval in the setting of brightness changes. We populate the database with darkened and lightened versions using $\theta_{photo} = [\delta, \delta, \delta]$, where $\delta > 0$ to mimic "daytime" and $\delta < 0$ to mimic "nighttime" versions of the images. We query both using $\Delta M(e(\mathbf{x}))$. Our steerable model retrieves other images in similar lighting settings as the query, whereas the invariant model retrieves the exact same nearest neighbors for the dark and light queries.

The results demonstrate the benefit of both equivariance and steerability: the equivariant embedding gives better results than the invariant embedding even for $e(g(x))$. Using the maps to further steer in suitable directions gives even more boosts in quality. Qualitatively, they also display the range of transformations and their parameters that our steerable equivariant respresentations generalize to. More results are shown in the Appendix (Figure 16).

### 4.2.2 Image Cropping/Zooming

In this experiment, we show that equivariant/steerable model preserves visual order for zooming data augmentation. Figure 5 shows the original image and a steered version (in embedding space). Each key image

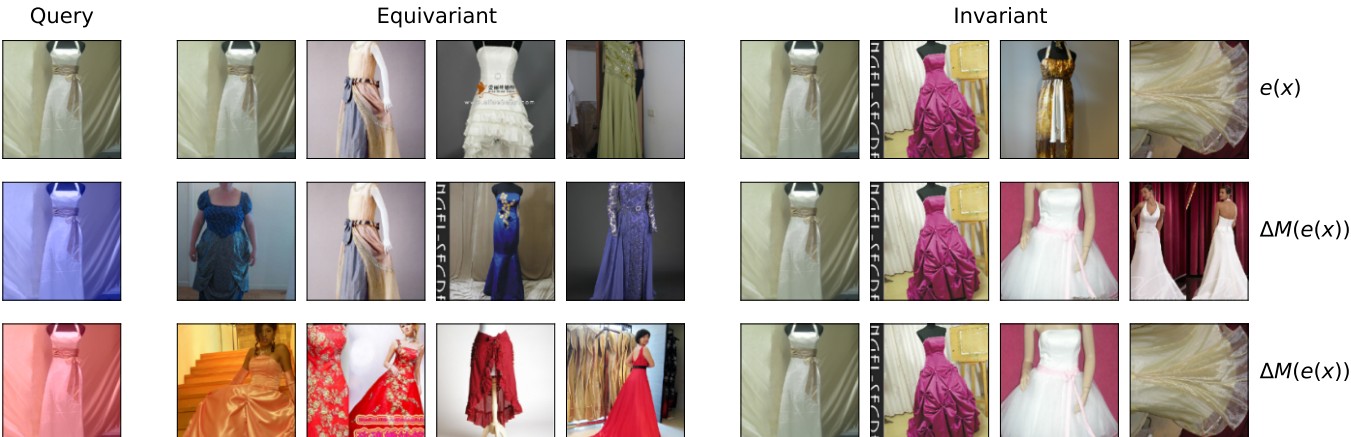

Figure 4: Color retrieval examples comparing our equivariant/steerable model to the invariant (standard) baseline. Steerable embeddings capture both visual and semantic relationships between the query and keys. The invariant model gives the same top nearest neighbors regardless of query color.

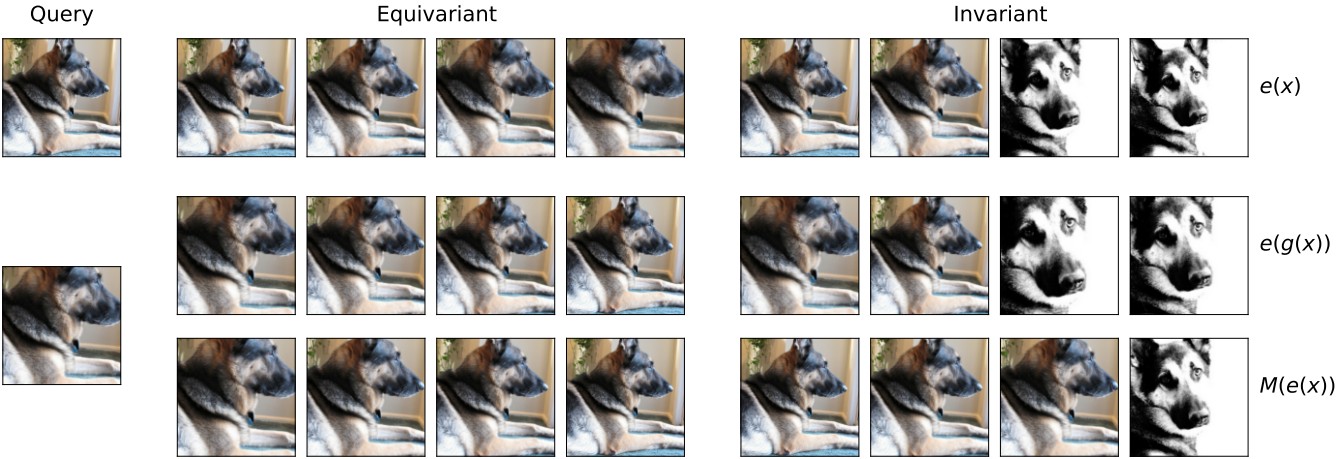

Figure 5: Examples of retrieval with crop/zoom data augmentation. See text for details. Equivariant model retrieves the same sample, ordered correctly by zoom level (e.g. see how the dog's head progressively gets exposed). Invariant model does not preserve the zoom ordering or retrieves other samples. See Appendix for other examples.

in the dataset consists of multiple zoomed versions of images from different classes. The equivariant model result maintains a sensible global ordering (retrieving samples from the same class) as well as local ordering (ordering the nearest neighbors according to the level of zooming). The invariant model does not preserve local ordering. For example, the equivariant model retrievals are correctly ordered by zoom level; whereas the invariant model retrievals orders them unpredictably.

### 4.2.3 Composed Augmentations

More complex sequences of augmentations are easily formed by applying the map functions sequentially. In Figure 6, we apply both photometric (color) and geometric (crop) augmentations in the database, and query using composed maps. The returned results respect both augmentations in a sensible manner (although there is no unique ordering). Note that the retrieved results respect high-level semantics (nearest neighbors belong

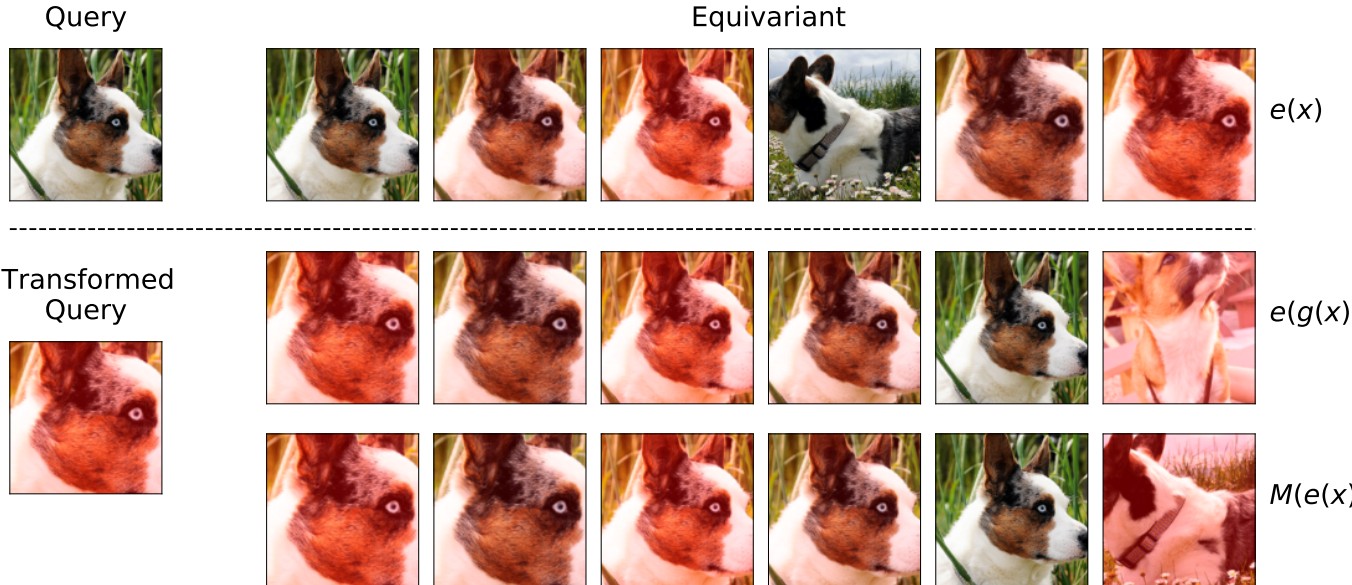

Figure 6: Image retrieval on composition of augmentations: color and crop transformations. We see that along each dimension (color or crop) ordering is preserved correctly.

to the same class) in addition to low-level attributes. We calculate MRR for this experiment as well, and report it in Table 2 (last column).

## 4.3 Transfer learning

While invariance to a particular transformation is useful for a particular dataset/task, it may hinder performance on another. Thus, we expect equivariant representations to perform better at transferring to downstream datasets than invariant representations. We test this by comparing the linear probe accuracy of both models on Oxford Flowers-102 (Nilsback & Zisserman, 2008), Caltech-101 (Fei-Fei et al., 2004), Oxford-IIIT Pets (Parkhi et al., 2012), and DTD (Fei-Fei et al., 2004) (see Table 1). We see that equivariant representations consistently achieve a higher accuracy.

## 4.4 Robustness and Out-of-Distribution Detection

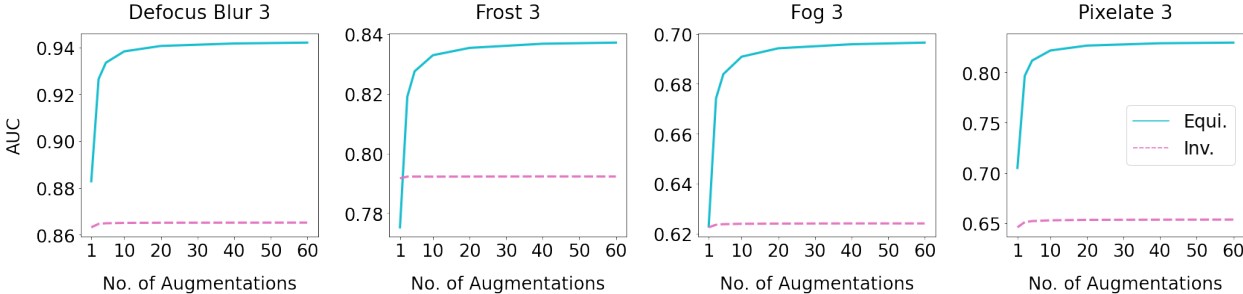

Figure 7: OOD for ImageNet (in-distribution) against 4 ImageNet-C corruptions (out-of-distribution). We use upto 60 crop augmentations. Equivariant AUC (latent) monotonically increases whereas invariant AUC (latent) stays nearly flat. Equivariant AUC's are 5%-15% better than that of invariant.

Invariance is commonly encouraged in model pre-training to improve robustness (Zheng et al., 2016; Geirhos et al., 2019; Rebuffi et al., 2021; Hendrycks et al., 2020). We test whether equivariance can then hurt in this setting vs invariance. We measure and compare the accuracy of the representations on various corruptions in

| Corruption | Equivariant | Invariant | Corruption | Equivariant | Invariant |
|---|---|---|---|---|---|
| Gaussian Noise | **37.53** | 33.89 | Snow | **33.01** | 30.94 |
| Shot Noise | **37.15** | 33.56 | Frost | **33.77** | 31.75 |
| Impulse Noise | **35.07** | 31.69 | Fog | **36.273** | 34.57 |
| Defocus Blur | **34.50** | 31.77 | Brightness | **39.26** | 37.69 |
| Glass Blur | **32.58** | 30.36 | Contrast | **40.00** | 38.30 |
| Motion Blur | **32.51** | 30.22 | Elastic Transform | **40.18** | 38.43 |
| Zoom Blur | **32.66** | 30.49 | Pixelate | **41.47** | 39.82 |
| JPEG Compression | **42.37** | 40.79 | | | |

Table 3: Accuracy of models on all the corruptions from the ImageNet-C (averaged across severities).

the ImageNet-C (Hendrycks & Dietterich, 2019) dataset in Table 3 , and find that the equivariant model is in fact suprisingly more robust on all the ImageNet-C corruptions. We also measure the mCE (lower is better) for both models and find that our model has an mCE of 0.81 as compared to the invariant model's 0.845.

Despite better robustness, there is a significant accuracy loss. In this case, we want our model to detect a sample with corruptions as out-of-distribution (OOD). Test-time data augmentation has enabled better performance on tasks such as detection of out-of-distribution, adversarial or misclassified samples and uncertainty estimation (Ayhan & Berens, 2018; Bahat & Shakhnarovich, 2020; Wang et al., 2019). These approaches are based on the hypothesis that in-distribution images tend to exhibit stable embeddings under certain image transformations. In contrast, OOD samples have larger variations. This difference in stability can be exploited to detect out-of-distribution samples. In existing work e.g. (Wang et al., 2019), multiple augmentations are applied to the input samples which are then forward propagated through the encoder. This leads to significant computational load since we typically need a large number of augmentations. This becomes increasingly impractical as the number of augmentations is increased. With our steerable model we can apply these augmentations directly in *embedding space*, leading to significant speedups. Applying 60 augmentations at input and then forward propagating them in a mini-batch takes **14.98** seconds. Conversely, forward propagating a single sample and applying 60 mappings in embedding space takes only **0.28** and **0.02** seconds per mini-batch respectively: a nearly 50× speedup. Applying 60 augmentations at input and then forward propagating them in a mini-batch takes **0.2263** seconds. Conversely, forward propagating a single sample and applying 60 mappings in embedding space takes only **0.0091** and **0.0030** seconds per mini-batch respectively: a nearly 50× speedup.

We perform OOD detection using ImageNet validation set as the in-distribution dataset and ImageNet-C as the OOD dataset. We use $M_{geo}$ to generate multiple augmentations in latent space for a given image, and compute AUC curves across augmentations (see Appendix A.7 for details). Results are shown in Figure 7 for 4 corruptions from ImageNet-C (remainder are presented in the Appendix).We see the clear benefit of applying latent augmentations for almost all corruptions and severity levels. We further see from Figure 7 that latent augmentations have an insignificant effect on the invariant model AUCs. Thus, these results demonstrate the benefit equivariant representations provide over invariant in test-time augmentations, and how steerability can be used to amplify these and obtain great computational speedups.

## 5 Conclusion

We have presented a method to *steer* equivariant representations in the direction of chosen data augmentations. To the best of our knowledge, ours is the first work to show a practical approach for general deep network architectures and training paradigms. We show the benefits of steerable equivariant embeddings in retrieval, robustness, transfer learning and OOD detection, with significant performance and computational improvements over the standard (invariant) model. Our method is simple to implement and adds negligible computational overhead at inference time. A limitation of our approach is that it requires the learning of new maps for every new data augmentation that we would like to steer.

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

# A    Appendix

## A.1    Training Details

Both the invariant and equivariant models use a ResNet-50 encoder trained for 250 epochs with a batch size of 4096, on 224x224 sized ImageNet images with AutoAugment (Cubuk et al., 2018) applied on the input to the cross-entropy loss. The remaining optimization details are as follows: SGD optimizer with 0.8 nesterov momentum, 0.1 learning rate with cosine decay warmed up over 12 epochs, $\alpha$=0.1 and $\beta$=0.1.

We plot the values of the equivariance metric, $\rho$, over the course of training in Figure 8.

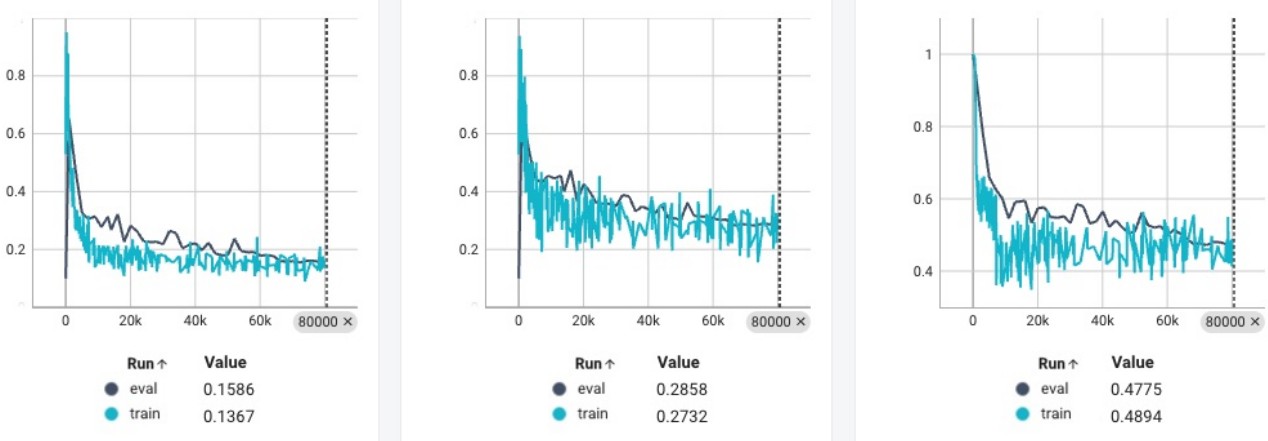

Figure 8: Values of $\rho_{photo}$ components over the course of training. Left: Numerator. Center: denominator. Right: $\rho_{photo}$

## A.2    Ablations on hyperparameters

In Table 4, we conduct ablations on hyperparameters $\alpha$ and $\beta$. We can see that both hyperparameters have sweet spots, above and below which the model either does not gain much equivariance, or it does but at

the cost of reduced accuracy. For the main paper, we empirically selected a model with hyperparameter values such that the cross-entropy accuracy is not adversely reduced w.r.t to the invariance los,; and the $\rho$ equivariance metric is reduced (lower is better) for all augmentations.

| $\alpha$ | Accuracy | $\rho_{geo} \downarrow$ | $\rho_{photo} \downarrow$ |
|---|---|---|---|
| 0.0 | 72.52 | 0.8752 | 0.8726 |
| 1e-3 | 74.2 | 0.8845 | 0.8914 |
| 0.01 | 69.51 | 0.8804 | 0.8658 |
| 0.1 | 75.51 | 0.884 | 0.8329 |
| **1.0** | 75.25 | 0.6482 | 0.4744 |
| 5.0 | 67.91 | 0.4722 | 0.3134 |

| $\beta$ | Accuracy | $\rho_{geo} \downarrow$ | $\rho_{photo} \downarrow$ |
|---|---|---|---|
| 0.05 | 75.01 | 0.5973 | 0.3239 |
| **0.1** | 75.25 | 0.6482 | 0.4744 |
| 0.2 | 74.63 | 0.8547 | 0.6345 |
| 0.5 | 74.44 | 0.8725 | 0.848 |
| 1.0 | 61.11 | 0.8757 | 0.8169 |

Table 4: Left: Ablation on $\alpha$, with $\beta$=0.1. Right: Ablation on $\beta$, with $\alpha$=1.0

## A.3 T-SNE

Visualising the nearest neighbours of transformed images, as we do in Section 4.2, can be thought of as a way to query the local structure of the latent space. An alternate way to analyse both the local and the global structure of the latent space is to perform dimensional reduction on it and visualise points directly in the reduced subspace. We apply multiple color transformations to a database of images from the ImageNet validation set, each progressively increasing in strength. We do these transformations both in image space, and directly in latent space using our steerable maps. We display an example of such transformations in Figure 9a. We then use t-SNE (van der Maaten & Hinton, 2008), a standard dimensional reduction technique for high-dimensional data, to obtain a 2-dimensional subspace for the set of original and transformed images. We plot these in Figure 9b.

We observe that the representations obtained by our method have a local linear structure in the low-dimensional subspace. This is thus more easily recoverable by our linear steerable maps, and the latent space transformations follow their input space counterparts closely. In the standard invariant representations, however, such a structure is missing.

## A.4 Comparison to ESSL

We compare against ESSL (Dangovski et al., 2022): an equivariant represenation learning method that enforces equivariance without explicitly defining equivalent latent-space transformations i.e. not enforcing steerability. For a fair comparison with our photometric augmentation, we train their base model with the color-inversion augmentation instead of the standard 4-fold rotation. We endow this model with a steerable map and train it with photometric augmentation with the encoder parameters frozen, similar to the invariant model in Section 4.

Results in Table 5 confirm that this model is more equivariant than the invariant baseline, but less so than our approach. in Figure 10, we observe that the transformed representations do not retrieve differently coloured nearest neighbours as well as our method. In fact, they are quite invariant to the colour change. However, using $\Delta M(e(\mathbf{x}))$, i.e. using steerability to 'push' the representations towards the transformation, retrieves better nearest neighbour results. Combined, these results demonstrate the superiority of our method in inducing equivariance in representations, and the second result further demonstrates the advantages of steerability even in equivariant representations trained without steerability.

## A.5 Rotation

We add rotation to the list of augmentations and measure model accuracy and $\rho$'s in Table 6. We see that the $\rho_{rot}$ (equivariance measure) is lower for our model than a standard (invariant) ResNet-50 in this case as well, and that the existing model accuracy and $\rho$ values for other augmentations are minimally affected. This confirms that our proposed method generalizes to rotations as well.

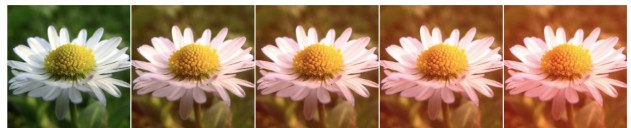

(a) Example of progressively severe colour augmentations.

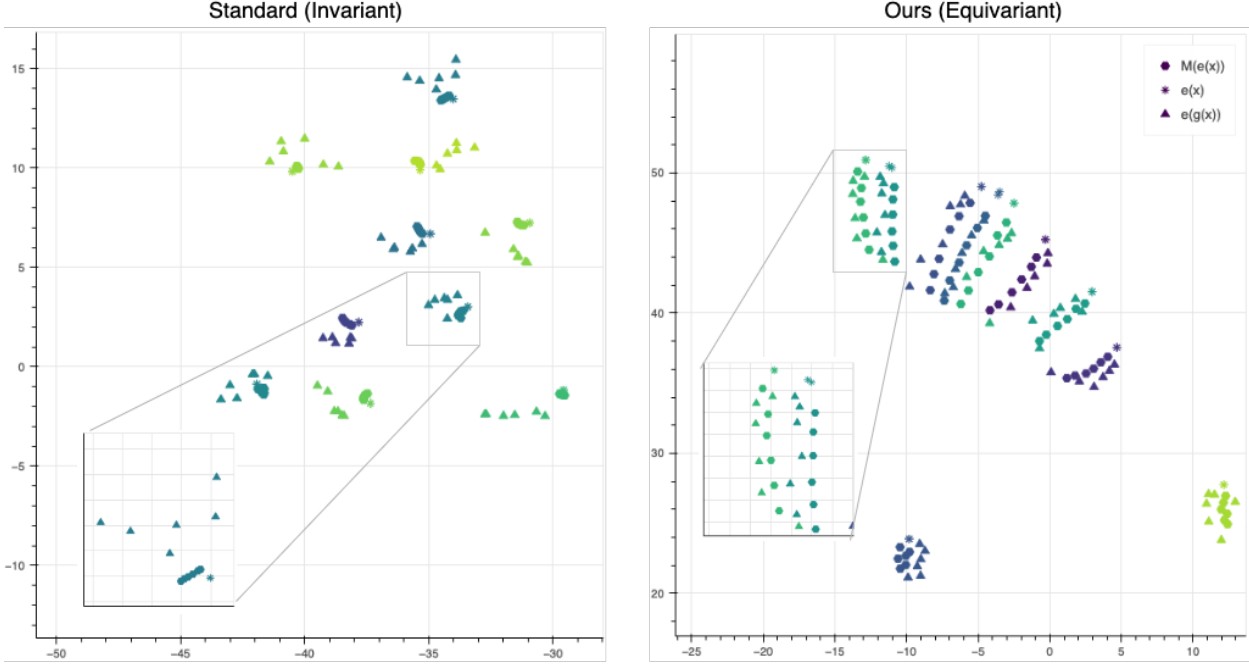

(b) Plots of original and progressively transformed images in a 2-D low-dimensional subspace obtained using t-SNE for invariant (left) and equivariant (right) embeddings.

Figure 9: t-SNE visualisations of original and transformed images.

| Model | $\rho_{photo} \downarrow$ | $MRR_{photo} \uparrow$ |
|---|---|---|
| Invariant (Standard) | 0.983 | 0.619 |
| Equivariant (E-SSL) | 0.6393 | 0.728 |
| Equivariant (Ours) | **0.474** | **0.974** |

Table 5: Equivariance measure ($\rho$, Eqn. 2) and MRR for photometric augmentation for ESSL.

| Model | Accuracy | $\rho_{geo} \downarrow$ | $\rho_{photo} \downarrow$ | $\rho_{rot} \downarrow$ |
|---|---|---|---|---|
| Invariant (Standard) | 75.17 | 0.8819 | 1.03 | 1.037 |
| Equivariant (Ours) | 75.18 | **0.535** | **0.502** | **0.498** |

Table 6: Accuracy and Equivariance measure (Eqn. 2) with rotation augmentation

## A.6 Robustness

In Table 7, we show the accuracy of both the models on ImageNet-C dataset (Hendrycks & Dietterich, 2019), on all 15 corruptions and 3 different severity levels. We can see that the equivariant model outperforms the invariant across the board, with better accuracy on all but 5 out of 45 data points.

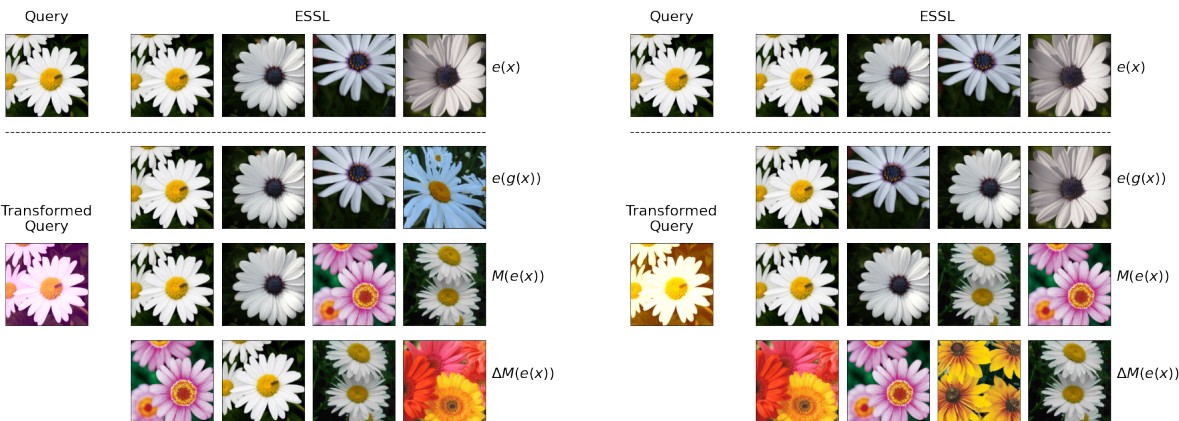

Figure 10: Nearest Neighbours for E-SSL.

|                        | Equivariant | | | Invariant | | |
|------------------------|---------|---------|---------|---------|---------|---------|
| Corruption  Severity   | 1       | 3       | 5       | 1       | 3       | 5       |
| Zoom Blur              | **62.61** | **39.88** | **8.44**  | 61.62   | 34.57   | 4.50    |
| Gaussian Noise         | **62.15** | **38.95** | **11.11** | 61.27   | 33.85   | 6.84    |
| JPEG Compression       | 50.23   | **34.93** | **7.95**  | **50.45** | 31.38   | 4.15    |
| Fog                    | **55.42** | **31.07** | **11.59** | 54.93   | 30.17   | 10.78   |
| Shot Noise             | **57.10** | **13.46** | **5.78**  | 56.73   | 12.91   | 5.25    |
| Impulse Noise          | **63.58** | **27.92** | **7.25**  | 61.82   | 24.00   | 5.65    |
| Defocus Blur           | **49.43** | **32.80** | **19.80** | 48.88   | 30.59   | 18.86   |
| Glass Blur             | **54.77** | **37.03** | **23.42** | 54.36   | 35.89   | 21.36   |
| Motion Blur            | **60.66** | **34.55** | **26.53** | 60.12   | 32.66   | 24.05   |
| Snow                   | 67.60   | 59.41   | 46.28   | **68.07** | **60.63** | **48.00** |
| Frost                  | **73.62** | **70.06** | **62.73** | 73.36   | 69.89   | 62.21   |
| Brightness             | **69.15** | **58.49** | **11.02** | 68.58   | 55.17   | 7.68    |
| Contrast               | **68.41** | **51.20** | **9.87**  | 68.01   | 48.11   | 8.14    |
| Elastic Transform      | **68.30** | **59.93** | **46.66** | 67.77   | 59.84   | 45.80   |
| Pixelate               | 62.42   | **57.54** | **43.91** | **62.87** | 57.44   | 41.07   |

Table 7: Accuracy of models on all the corruptions from the ImageNet-C with multiple severities.

### A.7   Out-of-Distribution Detection

Test-time Augmentation Details: Here we give details of how we perform test time augmentation. We use $M_{geo}$ / $M_{photo}$ to generate multiple augmentations in latent space for a given input image. We compute the geometric mean across the set of logits generated in this manner (for a given number of augmentations), and then use this average logit to compute softmax probabilities. The maximum softmax value is the confidence for this sample. We use these confidences across a set of ImageNet and ImageNet-C samples and probability threshold values to compute a PR curve, and measure the AUC of this PR curve. We repeat this for upto 60 augmentations, and plot the AUC values across the number of augmentations. In Figure 12, we provide more examples of OOD detection for all 15 corruptions from ImageNet-C (Hendrycks & Dietterich, 2019), with severity level 3. In 12 of the 15 cases, the equivariant latent outperforms invariant latent space in AUC on both photometric and geometric augmentations. We display similar plots on different severity levels with the geomentric augmentation in Figures 13, 14 and 15. Adding more number of augmentations may help to further improve performance on the equivariant model.

We also repeat the experiment above but by applying augmentations directly on the input images. We can only apply upto 8 input augmentations, as they use significantly more memory than latent augmentations.

The results are plotted in Figure 11. We see that (1) in general input augmentations do better than latent space augmentations but at a significantly higher speed, compute and memory cost; and (2) equivariant input augmentations always do better than for the invariant model. This shows the benefit of our equivariance promoting regularizer.

### A.8 Nearest Neighbor Retrieval

In Figures 16, 17, 18, 19, we show more examples of image retrieval with color, crop/zoom, composed-color-zoom, and brightness queries respectively across different classes. Qualitatively, this displays the range of transformations and their parameters that our steerable equivariance respresentations generalize to.

| $\beta$ | Accuracy | $\rho_{geo} \downarrow$ | $\rho_{photo} \downarrow$ | Flowers-102 $\uparrow$ | DTD $\uparrow$ |
|---------|----------|-------------------------|---------------------------|------------------------|----------------|
| 1e-3    | 75.17    | 0.8824                  | 0.8831                    | 83.59                  | 65.18          |
| 1e-2    | 75.23    | 0.8827                  | 0.8889                    | 83.26                  | 64.06          |
| 0.1     | 72.52    | 0.8752                  | 0.8726                    | 88.84                  | 66.52          |

Table 8: Ablation with $\alpha = 0, \beta > 0$

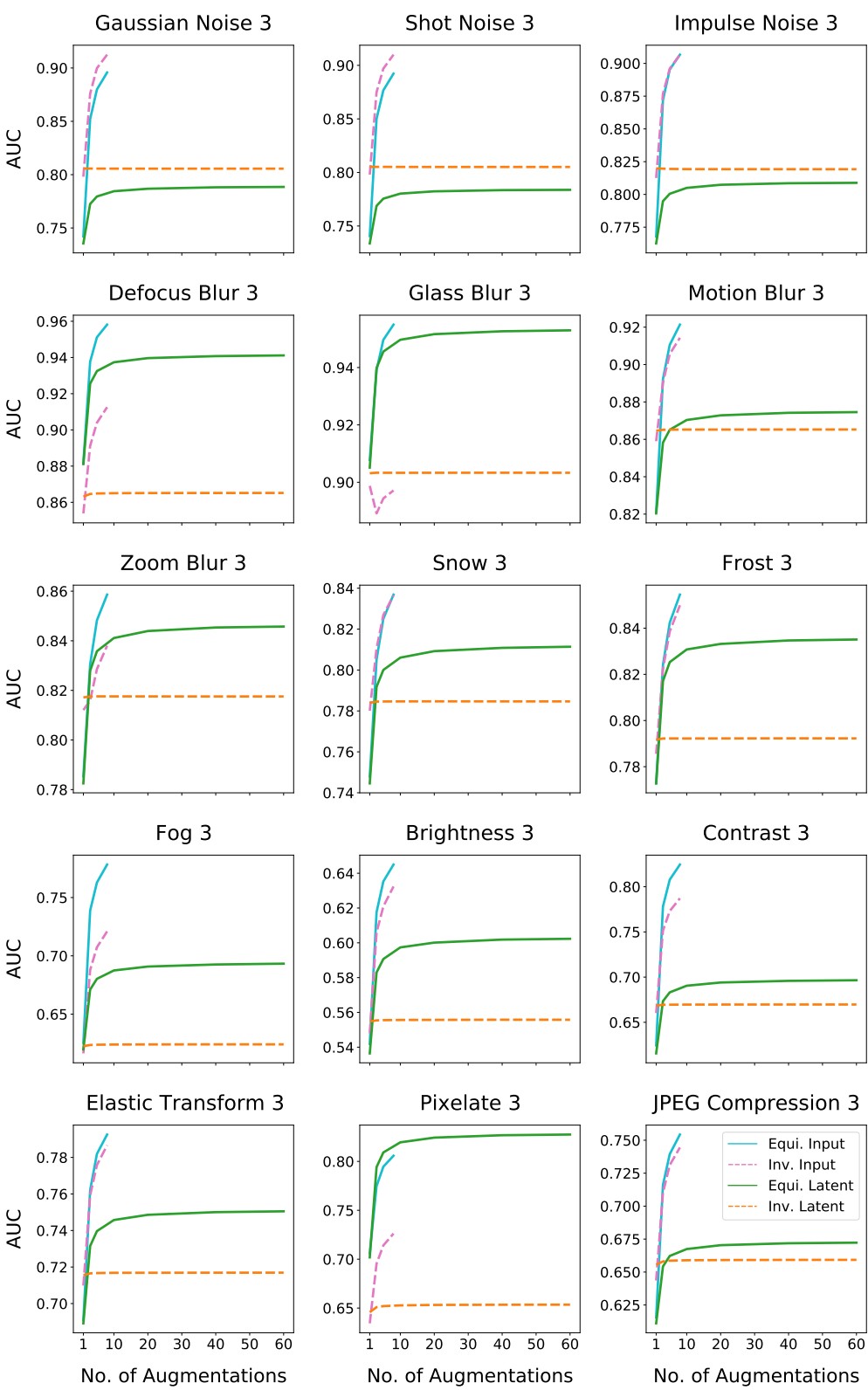

Figure 11: OOD Detection for ImageNet-C with geometric augmentations, applied either in input image or directly in latent embedding space.

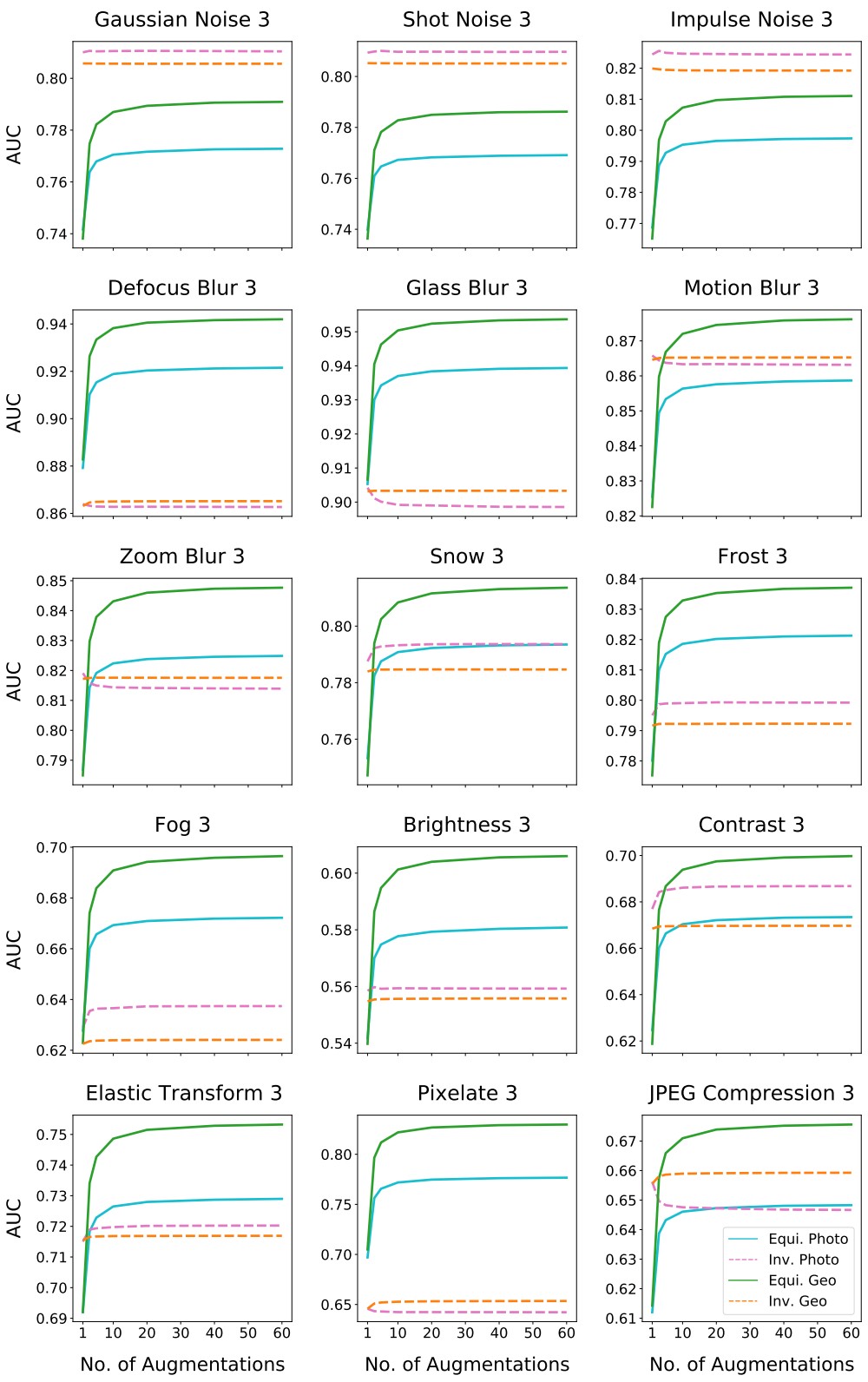

Figure 12: OOD Detection for ImageNet-C when both photometric and geometric augmentations are applied. We see that both augmentations lead to improved OOD performance.

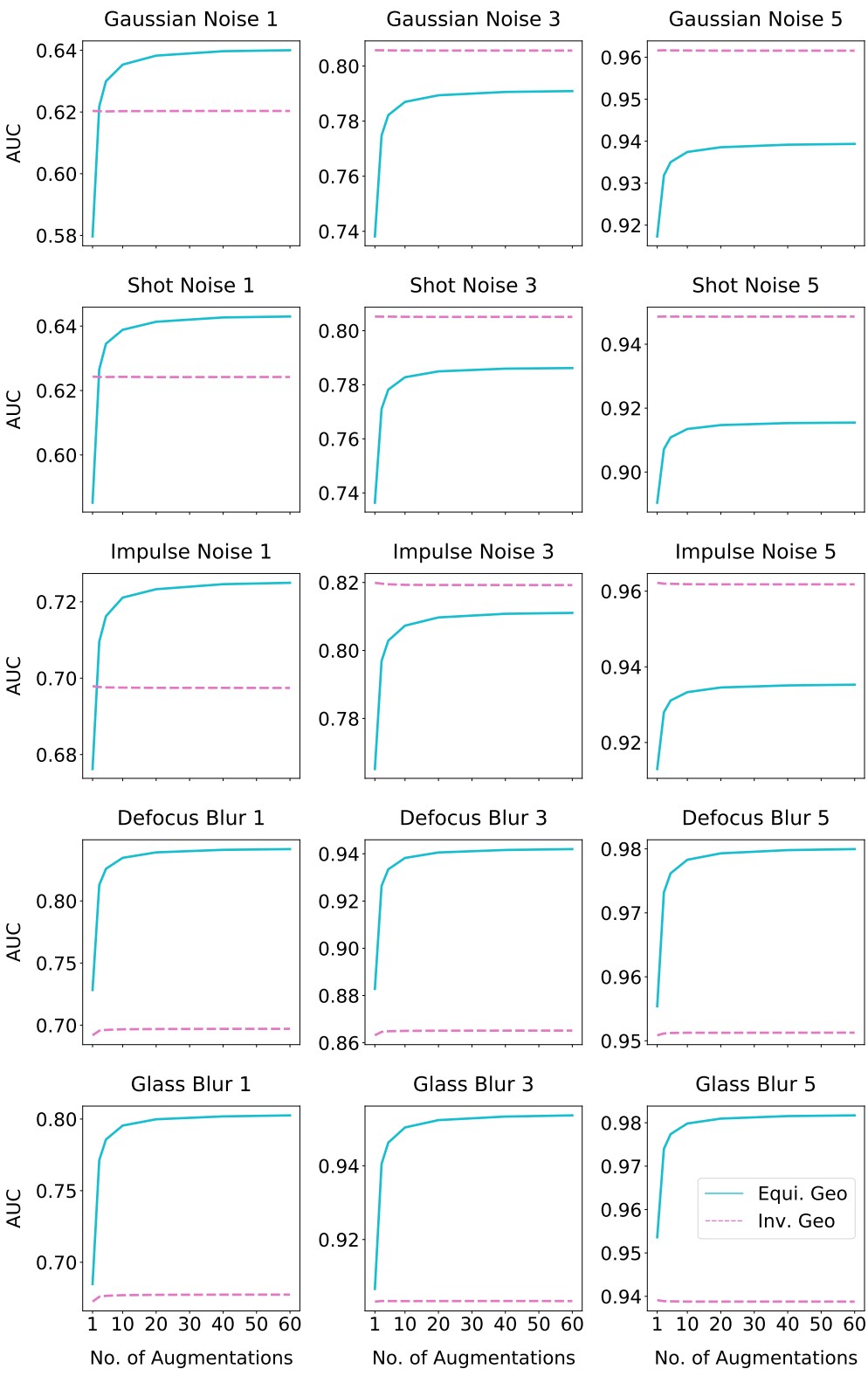

Figure 13: OOD Detection for ImageNet-C with geometric augmentations and multiple severity levels.

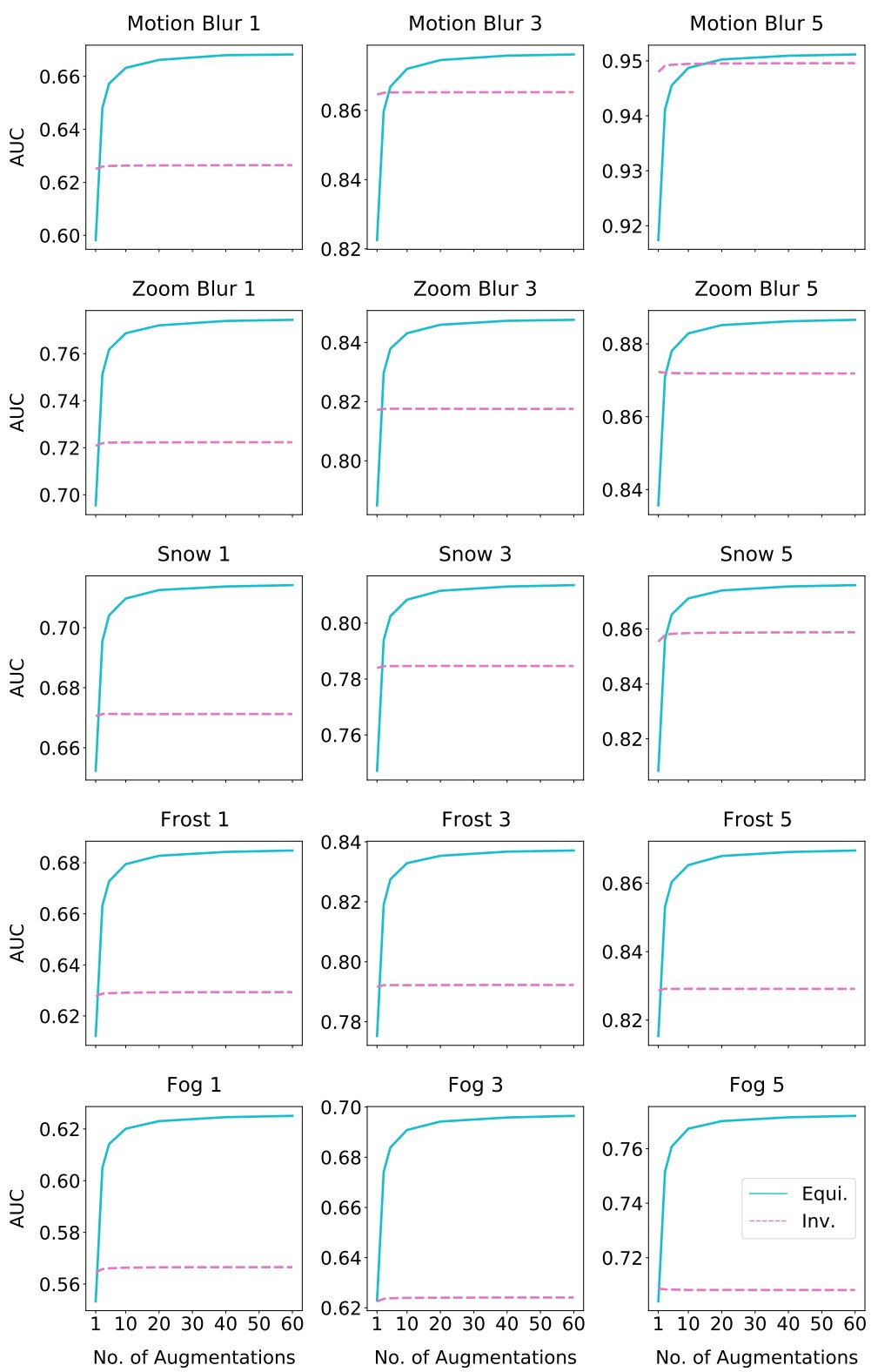

Figure 14: (contd.) OOD Detection for ImageNet-C with geometric augmentations and multiple severity levels.

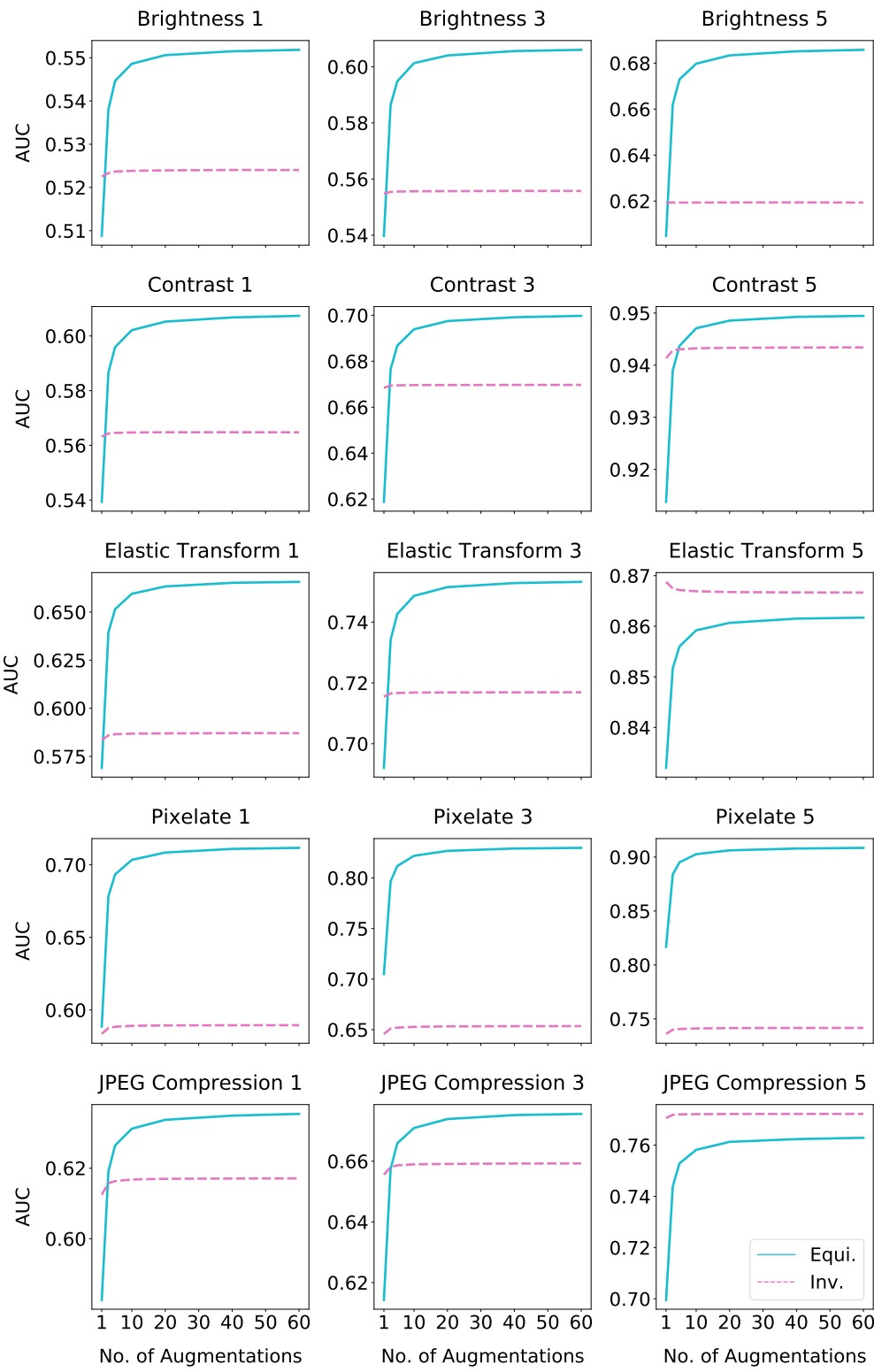

Figure 15: (contd.) OOD Detection for ImageNet-C with geometric augmentations and multiple severity levels.

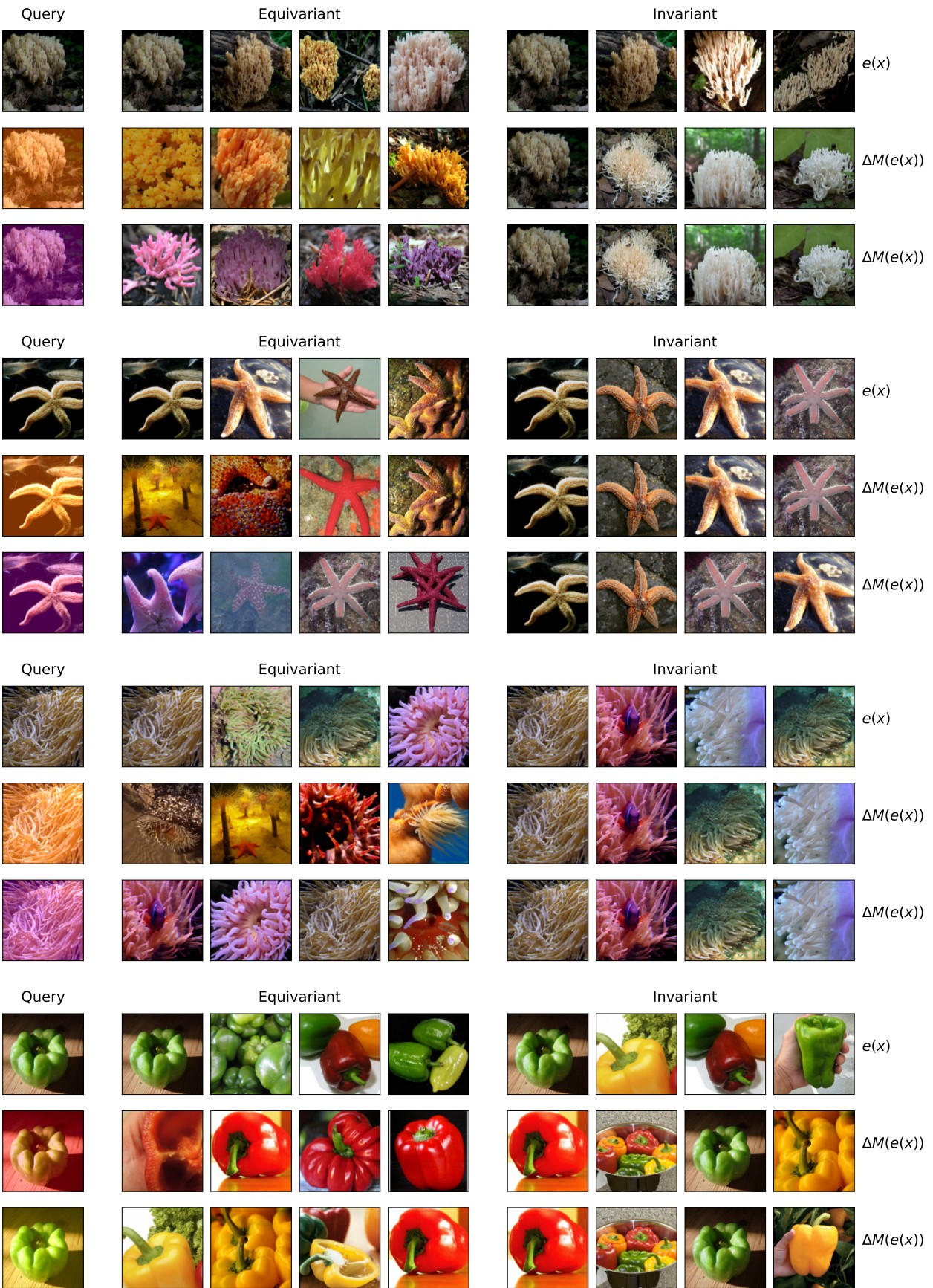

Figure 16: Image Color Retrieval Examples.

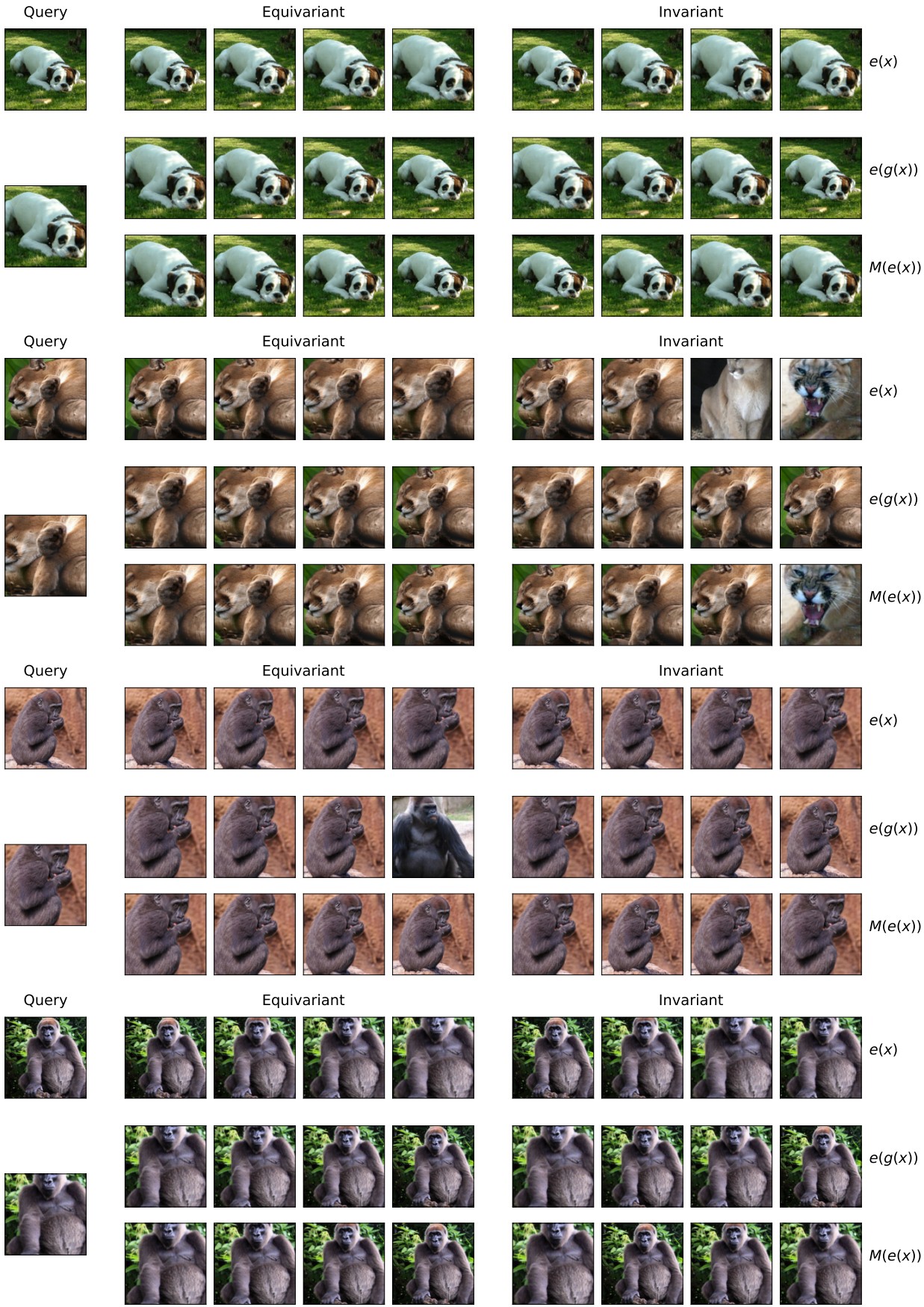

Figure 17: Image Crop/Zoom Retrieval Examples.

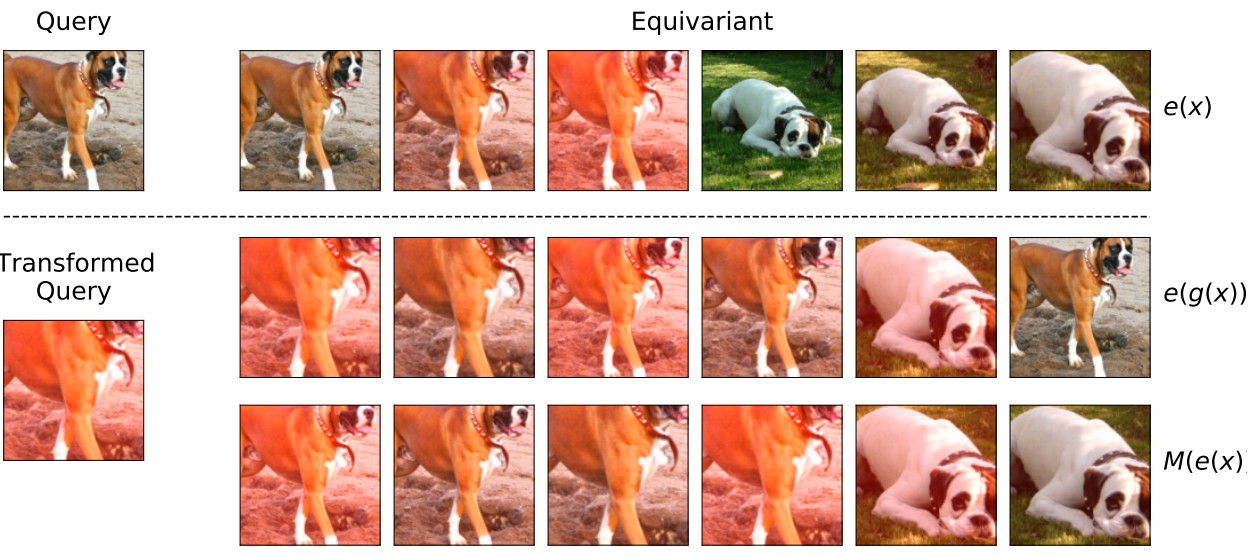

Figure 18: Image Color-Crop Composition Retrieval Examples.

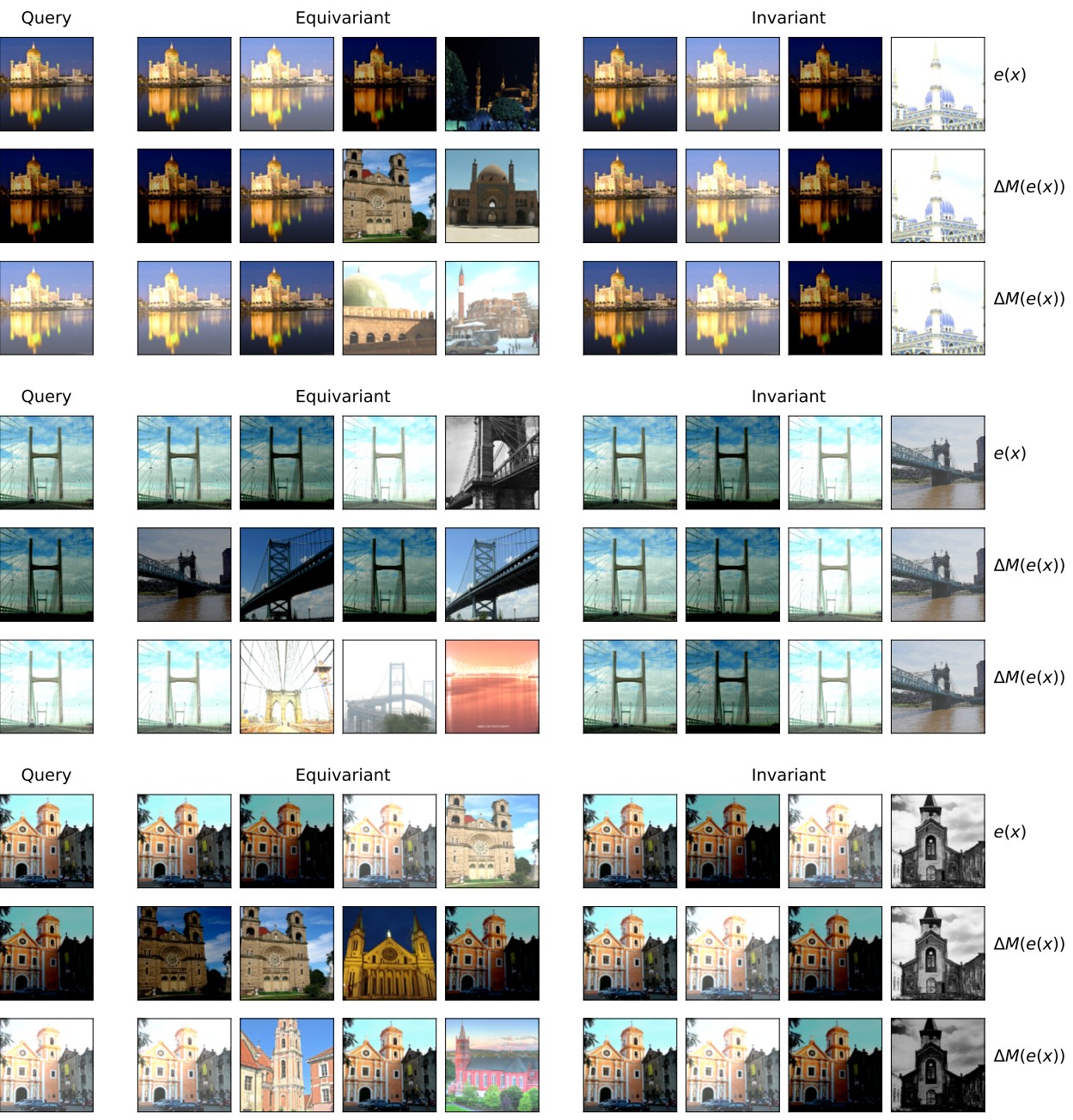

Figure 19: Image Brightness Retrieval Examples.

