# OpenReview forum: "Steerable Equivariant Representation Learning"
_TMLR — Rejected by TMLR_

### Review · Reviewer_xhKr · 2023-04-15

**Summary Of Contributions:**

The main contribution of this paper is to manipulate the equivariant representation learning in a steerable way. In such a way, the features in the embedding space can be manipulated directly with a learnable mapping function to formulate equivariance. The proposed equivariant learning method shows several benifits in the applications of retrieval, transfer learning, and robustness generalization/ood detection when comparing to the invariant learning method.

**Audience:**

Yes

**Broader Impact Concerns:**

No ethical concerns of this paper.

**Claims And Evidence:**

No

**Requested Changes:**

1. The experiments should present more accutately to verify the main contribution of this paper.

2. Some details of implementations of both methods and experiments should present more clear.

**Strengths And Weaknesses:**

## Strengths
1. Different from the previous equivariant representation learning, this paper proposes a more flexible way to manipulate the embedding features by a learnable transformation in the embedding space. By this, one could speed up the transformations and lead to computing benifits to downstream tasks.

2. Comparing to the invariant representation learning, this paper shows several benifits of equivarient representation learning in applications of retrieval, transfer learning, ood robustness, and ood detection tasks.

## Weaknesses
1. The empirical evidence to prove the advancement of the proposed method is not enough. Note that the equivariant representation learning is not new in community, and the main technical contribution of this paper is the proposed steerable way instead of the equivariant representation learning itself. Therefore, the empirical study should focus on more to verify the effectiveness of "steerable way". However, throughout the experiments, the proposed method is only comparing to that of invariant representation learning. So, the benifits of the proposed method are not fully verified when comparing to the previous equivarient representation learning.

2. Some parts of descriptions of the method and experiment are not very clear. E.g., How the $M_\theta$ is constructed in the embedding space? and how the test-time augmentation is used in the OOD detection task? In the retrieval experiment, what the $M_\theta$ is in the invariant representation learning? If the $M$ is the identical function in the invariant version, then why we could get differnet retrival results in those figures of $M(e(x))$. Also, due to the importance of hyperparameters, how the model selection is conducted is not clear.

---

> ### Author Response · Authors · 2023-05-26
> **Response to reviewer xhKr**
>
> We thank the reviewer for their constructive feedback which helps to considerably strengthen the paper. Please find some detailed responses to review points below, as well as a revised manuscript with tracked changes.
> - Response to “ Note that the equivariant representation learning is not new in community, and the main technical contribution of this paper is the proposed steerable way instead of the equivariant representation learning itself. Therefore, the empirical study should focus on more to verify the effectiveness of "steerable way". However, throughout the experiments, the proposed method is only comparing to that of invariant representation learning.”: \
> \
> We agree that equivariant learning is not new in the community, and that ways of adding equivariance can take many forms. We would, however, like to reiterate that we do note that our proposed method has other differences and benefits over previous work beyond steerability, and we also demonstrate the added effectiveness of steerability over equivariance. \
> Specifically, architecture-based techniques, as noted in the introduction, work only for a limited set of transformations and have to be hand-crafted for each one. The techniques of [1, 2], are limited to the setting of contrastive learning. Our method is much more general, both in the set of transformations and architectures / training paradigms it can be adapted to.
> Also, we would like to point out that while equivariance is already studied for improving classification, its benefits had not previously been demonstrated in the nearest neighbour retrieval and OOD detection settings. \
>  Furthermore, we demonstrate that steerability improves performance further in both these cases. For example, we see in Figure 2 that exploiting steerable maps gives even better nearest neighbour results, and in section 4.4 that they also lead to 50x speedup in OOD detection.
> We thank the reviewer for pointing this out, we have added a summary of our contributions at the end of the introduction to make these points clearer in the paper. \
> \
> However, to further prove the benefits of our approach in improving equivariance, we have added comparisons to Dangovsky et al [1]. Since their method does not lend itself immediately to steerability via learnable maps, we endow their model with steerable maps and train them with the encoder parameters frozen. This is similar to how we add maps to the invariant baseline in Section 4. We refer the reviewer to Section A.4 for these added comparisons and results. To summarize, they demonstrate the superiority of our method in inducing equivariance in representations, and also the advantages of steerability even on frozen equivariant representations trained without steerability.

---

> > ### Author Response · Authors · 2023-05-26
> > **Response to reviewer xhKr Contd.**
> >
> >
> > - Response to “Some parts of descriptions of the method and experiment are not very clear”: We apologize for the confusion in these descriptions. We clarify these details:
> >   - "How the M(theta) is constructed in the embedding space?" \
> >     Section 3.2 details how the map M is constructed.
> >   - "and how the test-time augmentation is used in the OOD detection task? " \
> >     Section A.7 contains details of how the multiple augmentations are combined for OOD detection at test-time.
> >   - "In the retrieval experiment, what the M(theta) is in the invariant representation learning?"\
> >     Section 4’s introduction contains details of how the invariant model is also endowed with maps trained on a frozen representation, to facilitate comparison in a steerable setting.
> >    - "If the M is the identical function in the invariant version, then why we could get differnet retrival results in those figures of M(e(x))." \
> >      Because of complex optimization, it is reasonable that the representations are not perfectly invariant. This is reflected in Table 1, where a perfectly invariant representation would have achieved $rho_g = 1$. Since M then is not perfectly identity, the results are not identical. However, please note that the results in Figure 2 are very similar for all rows of the standard representation (with 3 out of 4 neighbours shared), whereas for our method there is a lot more diversity. This confirms that the standard representations are much more invariant to the applied transformations than those obtained by our method.
> > We add that our results further showcase that imposing even some minor degree of steerable equivariance is significantly beneficial. Indeed, even if we do not achieve perfect equivariance for the reasons outlined above, the functional gains are important which speaks to the strength or our approach.
> >
> >   - "Also, due to the importance of hyperparameters, how the model selection is conducted is not clear." \
> >    We trained both models with the same hyperparameters for a fair comparison. We refer the reviewer to Section 4’s introduction and Section A.1 and A.2 for details
> >
> > References:
> > 1. Rumen Dangovski, Li Jing, Charlotte Loh, Seungwook Han, Akash Srivastava, Brian Cheung, Pulkit Agrawal, and Marin Soljačić. Equivariant contrastive learning, 2022
> > 2. Tete Xiao, Xiaolong Wang, Alexei A Efros, and Trevor Darrell. What should not be contrastive in contrastive learning

---

> > > ### Comment · Reviewer_xhKr · 2023-05-30
> > >
> > > My concerns have been addressed, and thanks for the detailed responses.

---

### Review · Reviewer_Q7EF · 2023-04-19

**Summary Of Contributions:**

**Contributions**:

1. This work introduces an equivariance-promoting regularizer (based on the definition of equivariance) and a uniformity regularizer (previously investigated) that together foster the development of learned representations that are equivariant to input transformations.
2. The work empirically demonstrates that the learned equivariant representations improve performance in image retrieval, transfer learning, and robustness. The proposed methodology also offers significant test-time acceleration for augmentations, as it allows for direct transformations of the embeddings to augment the data.

**Audience:**

Yes

**Broader Impact Concerns:**

I have no ethical concerns.

**Claims And Evidence:**

No

**Requested Changes:**

1. The experimental results demonstrate that the proposed regularization terms improve performance on downstream tasks. However, it remains inconclusive whether this improvement is due to the learned representation being equivariant. To provide a more comprehensive analysis of equivariance, I suggest the following investigations:
- The current measurement of equivariance is also utilized in training the model, so it is quite expected that lower numbers will be obtained when employing it as a metric. I recommend using an alternative equivariant measurement, such as the [Lie Derivative](https://arxiv.org/pdf/2210.02984.pdf), to better analyze the learned equivariance.
- Additionally, it would be beneficial to visually examine the learned equivariant representations. One possible approach (though you may conceive a better one) is to use the proposed regularization to train an autoencoding architecture in which the learned embeddings are used to reconstruct the inputs. By applying the learned transformation in the embedding space, you can observe whether the reconstructed outputs undergo the corresponding transformation.
2. It would be good if the authors could be more precise about terminology and scientific statements, and clearly define them when needed. Many statements in the main text are a bit confusing and difficult to follow. To give a few examples:
- There is no clear definition of steerability. The authors state that "It is possible to make representations equivariant but not steerable, by not recovering the M's". I cannot understand this sentence. I guess the concept of steerability is important for this paper (as it appears in the title). Then it is worthwhile to formally define it.
- "An equivariant embedding changes (smoothly) with respect to changes at the input". Change is a vague term. Do you mean transformation? Generally, it is good to avoid oral sentences.
- "Using a dense layer" Can you be precise if it is a linear mapping or with nonlinearity?
There are many other places like this including some minor typos. It would be worthwhile for authors to go through the text carefully and improve the accuracy of their statements, and formally define the terms if necessary.
3. The current presentation makes it challenging to place the authors' work in context and discern how it differs from previous research. The authors state that "xxxx propose more propose more flexible approaches to allow arbitrary input transformations to be represented at the embedding, for the self-supervised setting. However, a key distinction between these works and ours is that we parameterize the transformations in latent space, allowing for steering." Upon reading this sentence, it is difficult to understand how representing transformations in the embedding space is different from parameterizing transformations in the latent space. As it stands, the introduction and related work sections do not adequately convey the authors' contributions in context.

**Strengths And Weaknesses:**

**Strengths**:

1. The empirical results are interesting because they seem to confirm that learned equivariant representations are useful for many downstream tasks.
2. The proposed regularization terms are simple to use and are agnostic to architectures.

**Weaknesses**:

1. The authors may not be careful enough with the use of terminology and the accuracy of their statements, which often leads to confusion.
2. The investigation of equivariance and its potential role in driving performance improvements is insufficiently explored. (see requested changes below).
3. While the authors include background and related work, the current writing is somewhat confusing. Understanding their contributions in context is difficult without delving into all the referenced literature.

---

> ### Author Response · Authors · 2023-05-26
> **Response to reviewer Q7EF**
>
> We thank the reviewer for their constructive feedback which significantly helped improve the paper. Please find some detailed responses to review points below, as well as a revised manuscript with tracked changes. We respond to comments below:
> - Response to “The current measurement of equivariance is also utilized in training the model, so it is quite expected that lower numbers will be obtained when employing it as a metric.”: \
>   We respectfully disagree with the statement that the current measurement of equivariance is used for training the model also. The metric $\rho_a$ is normalized, and it is not obvious that training using our regularizer will not cause representation collapse or trivial invariant solutions. We also provide other quantitative and qualitative experiments to show more equivariance is achieved using our method.
> - We thank the reviewer for pointing us to the work on Lie Derivatives for analysing equivariance. We would like to point out that $\rho_a$ is in fact equivalent to the Expected Group Sample Equivariance metric in Section C.3 of that paper. The authors show in Section 6 that this metric is correlated with and shows similar trends as the Lie Derivative. As a result, we argue that $\rho_a$ is also expected to have similar trends as the Lie Derivative. We added a note about this interesting connection in the text (see Section 3.1), and thank the reviewer once more for pointing this out.
>
> - Response to “One possible approach (though you may conceive a better one) is to use the proposed regularization to train an autoencoding architecture in which the learned embeddings are used to reconstruct the inputs”: \
>   While we agree that visual analysis of the representations through generative methods would be interesting, we explicitly avoid this approach because it introduces additional confounding variables in the form of the decoder. Indeed, the simulataneous training of a decoder prevents the isolation of the the effect of equivariance, as it is possible for the decoder itself to be invariant to the transformations of the latent space. As such, the visualizations using Nearest Neighbour approaches, which are admittedly indirect,  allow us to query the structure of the latent space without confounders. We feel this is an important distinction and would be happy to discuss further with the reviewer if they feel otherwise. \
>   Nevertheless, we appreciate the reveiwer’s demand for additional evaluations and we propose alternate visualizations using t-SNE. We refer to the reviewer to section A.3 for these added visualizations. In summary, we find that our steerable equivariance imposes quite structured representations to emerge, with smooth and regular layout of augmented inputs, as expected. In contrast, this is not the case with invariant representations.
>
>
> - Response to “It would be good if the authors could be more precise about terminology and scientific statements, and clearly define them when needed”: \
>   We apologise for the lack of clarity in these statements. We have rewritten the introduction to add a more precise definition of equivariance and steerability, and to contrast better against previous work. We have also amended the details of the architecture. We refer the reviewer to purple-coloured sections in the Introduction and other text for these changes.

---

### Review · Reviewer_PSUN · 2023-05-04

**Summary Of Contributions:**

This work proposed a regularization term in the loss function to achieve steerable equivariant representation in a supervised setting. The resulting model shows some qualitative steering during retrieval and quantitative performance gain in OOD benchmarks. However, I can not interpret if steerability or equivariance has been achieved.

**Audience:**

Yes

**Broader Impact Concerns:**

I do not have an opinion on this.

**Claims And Evidence:**

No

**Requested Changes:**

The following are some suggestions to make the paper more convincing, but it also involves my personal bias.

1. To show on simple example that steerability and equivariance can be exactly achieved.

2. Use better natural transform space in benchmark datasets to demonstrate steerability and equivariance. The color jittering and random resizing crops are not ideal for showing these properties.

3. Use more detailed visualization to show steerability and equivariance have been achieved. The current visualization is very hard to interpret.

4. To do more ablation studies to show the contribution of each introduced term. There are only two terms, so this should be quite convenient to do.



**Strengths And Weaknesses:**

Equivariance and steerability are important properties for representation learning. The results look somewhat promising.

However, I don't understand what was achieved in this paper. In my opinion, neither equivariance nor steerability has really been demonstrated in the current paper. The OOD results are interesting, but they can also be the result of the other uniformity term introduced.

The authors have not demonstrated in simple cases that equivariance or steerability has been achieved. With real datasets, the equivariance term is far from 0, which makes the quantitative measurement hard to interpret. Further, the qualitative visualization is also unconvincing in showing equivariance or steerability.

The OOD performance can be achieved with many different techniques. Essentially, any anti-collapse regularization shall provide some effect on OOD performance.

Parameterizing random resizing crops and color jittering is not the ideal space to demonstrate steerability or equivariance. The author may want to find more convincing ways to show steerability, e.g., with the natural geometric transformation of the object, like 3D rotations or so.

---

> ### Author Response · Authors · 2023-05-26
> **Response to reviewer PSUN**
>
> We thank the reviewer for their feedback. Please find some detailed responses to review points below, as well as a revised manuscript with tracked changes.
> - In response to “With real datasets, the equivariance term is far from 0, which makes the quantitative measurement hard to interpret. Further, the qualitative visualization is also unconvincing in showing equivariance or steerability.”\
>   We acknowledge the equivariance term is far from 0, which is expected when a network is optimized for a loss containing multiple terms such as classification loss, in addition to equivariance regularization. We have provided comparisons against an invariant baseline to make the quantitative measurements interpretable. For example, Table 1 demonstrates that $\rho_a$ decreases significantly compared to an invariant representation.
>   We argue that the qualitative and quantitative comparisons in figures 1, 3, 4, and tables 1 and 2 are demonstrating both the effects of equivariance and steerability. The standard representation in all figures does not respond to the transformations of inputs, pointing to invariance. Our representation on the other hand gives reasonable results for many different parameters and settings, indicating equivariance. All results with $M(e(x))$ correspond to latent transformations achieved with steerability.
>   We add that our results further showcase that imposing even some degree of steerable equivariance is significantly beneficial. Indeed, even if we do not achieve perfect equivariance for the reasons outlined above, the functional gains are important which speaks to the strength or our approach.
> - In response to “Use better natural transform space in benchmark datasets to demonstrate steerability and equivariance. The color jittering and random resizing crops are not ideal for showing these properties.” \
>   We respectfully disagree that the given transformations are not ideal for demonstrating equivariance and steerability. These are standard data augmentations, and are used also by previous work [1, 2]. Nearest-neighbour experiments to retrieve coloured or zoomed versions of images have real-world applications in image search and are quite relevant for image-related products.
> Moreover, our method, as well as others, rely on applying the transformation in input space first for training. This is why methods studying invariance and equivariance use color jittering, crops, rotations etc. as transformations. Natural space transformations like 3D rotation are infeasible to apply to existing images, and are hence not used as data augmentations in standard literature.
> - In response to “The OOD performance can be achieved with many different techniques.” \
>   We agree that different paths to OOD are possible, however we explore equivariance in this paper, and therefore demonstrate the effectiveness of this approach. Future work might aim to combine our approach with other existing good practices and we are excited about the prospects.
> - In response to “The current visualization is very hard to interpret.”: \
>   We apologize if our visualization was difficult to interpret.. We have improved figure captions as well as added figures (see appendix and other experiments) in an effort to clarify. Specifically, we refer the reviewer to Section 4.2 for more details, and to the captions of figures 1 and 3 (amended to add more details). We are happy to clarify any specific questions about the visualizations. We also refer the reviewer to Section A.4 for added t-SNE based visualizations.
> - In response to "To do more ablation studies to show the contribution of each introduced term." \
>   We have conducted and presented ablations in Section A.2 in the appendix. We add a note pointing to this in Section 4.
>
>
> References:
>
> 1. Rumen Dangovski, Li Jing, Charlotte Loh, Seungwook Han, Akash Srivastava, Brian Cheung, Pulkit Agrawal, and Marin Soljačić. Equivariant contrastive learning, 2022
> 2. Tete Xiao, Xiaolong Wang, Alexei A Efros, and Trevor Darrell. What should not be contrastive in contrastive learning

---

> > ### Comment · Reviewer_PSUN · 2023-06-21
> > **Claiming Big Without Actually Doing the Work ..**
> >
> > 1. First of all, the title "Steerable Equivariant Representation Learning" has two very nice words "steerable" and "equivariant". However, the results of this work can not hold up such nice buzzwords. After all, the authors just did a regularization and simple parameterization of the data augmentation (resized cropping & color jittering). The authors may want to rename the work properly to respect readers' time. Otherwise, a lot of interested readers caught by this nice title may find such a catchy name flawed.
> >
> > 2. The biggest issue is that the authors should deliver the results to support the claims!
> > 2.1 First, can you actually systematically show that the baseline embedding (not projection) space is invariant? What does "stays nearly constant" mean? If it is not constant, then potentially, one can train a predictor based on the augmentation parameters to predict the embedding. Have the authors tried this simple strategy?
> > 2.2 "as long as a transformation is parameterized in input space, we can train a mapping function in embedding space to mimic that transformation" Have the authors actually gone beyond the data augmentation transform and showed this method works well?
> > 2.3 Even with the color augmentation, the visualization results are very hard to interpret about "equivariance" and "steerability".
> > 2.4 Most of the results are not actually talking about steerability or equivariance. They are some special transfer settings, OOD, etc. Why don't the authors study steerability and equivariance systematically and deeply first? It will make this work much more attractive.
> >
> > 3. The introduction of the equivariance and steerability is handwavy and sometimes wrong. "In other words, the actions of g in input space commutes with the action of M in embedding space" "In the map, M is well-behaved .." "It has been shown that pre-trained embeddings often accommodate linear vector operations .." "When this response is trivially the identity function, an embedding is invariant to the transformation, otherwise, it is equivariant .." What is the desired property of "equivariance" in this work? Please definite it clearly and show you can achieve it.
> >
> > If the authors want to claim this big, please actually do that work. Or if the authors do not have the capacity to do the work, please consider revising the title and claims seriously. Further, I highly suggest the authors to carefully revise the writing .. I suggest at least a major revision for this work at this point.

---

> > > ### Author Response · Authors · 2023-06-21
> > > **Response to "Claiming Big Without Actually Doing the Work"**
> > >
> > > We thank the reviewer for this response. Please find some rebuttal points below.
> > >
> > > 1. We are uncertain about how to quantify the extent to which a body of work holds up to buzzwords, but we hope that a more nuanced title could alleviate the reviewer’s concerns. We propose a revised title along the lines of “Towards equivariant representation learning by promoting steerability”. While the reviewer states that our method is “ just [ ] a regularization and simple parameterization of the data augmentation”, we respectfully disagree. Introducing novel loss terms which act as regularizers is one of the few ways one can influence optimization, and can have highly non-trivial effects. Our proposed method introduces a simple, yet novel term that promotes equivariance. We see this simplicity as an asset as it can be easily implemented and applied in a wide range of conditions. We are unaware of any prior work outlining the advantages of such an intervention on optimization, and argue that it constitutes a valuable contribution to the literature seeking to promote equivariance in representation learning. Finally, we respectfully point out that our method is general and can be applied to a number of transformation domains, and that we choose to demonstrate its effectiveness on augmentation which include resized cropping & color jittering, as these are important features used in practice. We refer to our previous rebuttal for further details about our choice of numerical experiments.
> > >
> > > 2.
> > >     1. In response to “What does "stays nearly constant" mean? If it is not constant, then potentially, one can train a predictor based on the augmentation parameters to predict the embedding. Have the authors tried this simple strategy? ”. When referring to the phrase "stays nearly constant", we assume the reviewer points to line 10 in the very first paragraph of the introduction where we write “For invariant embeddings, the (output) embedding stays nearly constant for all transformations of a sample (e.g. geometric or photometric transformations of the input). Invariance is desirable for tasks where the transformation is a nuisance variable (Lyle et al., 2020). However, prior work shows that it can lead to poor performance on tasks where sensitivity to transformations is desirable (Dangovski et al., 2022; Xiao et al., 2021).” We note that this is meant to introduce the difference between invariance and equivariance as well as to motivate our pursuit of equivariance. To be invariant to a transformation means that embeddings of an input and its transformed version are mapped to similar (ideally equal) points in embedding space. Learned embeddings are usually not perfectly invariant (or equivariant) so small differences inevitably remain.
> > > We are unsure as to what the reviewer means by “ one can train a predictor based on the augmentation parameters to predict the embedding.” Our $M$ mappings are predictors of the transformed embedding, based on the augmentation parameters $\theta$.
> > >
> > >     2.  In response to "”as long as a transformation is parameterized in input space, we can train a mapping function in embedding space to mimic that transformation" Have the authors actually gone beyond the data augmentation transform and showed this method works well?”. We are confused by this comment. What the reviewer refers to as “data augmentation” are examples of transformations that impact practical applications such as image classification. We choose a variety of such transformations to illustrate the validity of our method but also note that it is quite general and can be applied beyond our chosen examples. An exhaustive exploration of all potential applications of this method is beyond the scope of this paper. If the reviewer has a specific experiment in mind, we would be open to discuss its inclusion but note that we already have several distinct demonstrations.
> > >
> > >     3. In response to “Even with the color augmentation, the visualization results are very hard to interpret about "equivariance" and "steerability".” We note that the reviewer made similar general remarks in the initial review and that we made substantial efforts to improve interpretability in our previous rebuttal where we outlined all changes made in the manuscript and made such changes easy to track in the revision. We would kindly ask if these changes addressed the reviewer’s concerns and if not, why?

---

> > > ### Author Response · Authors · 2023-06-21
> > > **Contd. Response to "Claiming Big Without Actually Doing the Work"**
> > >
> > > 2.
> > >     4. In response to “Most of the results are not actually talking about steerability or equivariance. They are some special transfer settings, OOD, etc. Why don't the authors study steerability and equivariance systematically and deeply first? It will make this work much more attractive.” Our results go as follows: we present a novel method to promote equivariance representations. Using illustrative examples based in image processing and classification, we measure the impact of this method with a metric designed to evaluate equivariance. Finally, we evaluate the benefits of our method promoting equivariance on classification tasks which include OOD and transfer settings. In light of this, we are unsure about what the reviewer means by “study [..] systematically and deeply”. If the reviewer makes such big requests, we kindly ask that they at least provide some indication of what they mean.
> > >
> > > 3. We respectfully ask the reviewer to point out what part of our introduction of equivariance is wrong. We also note that in the second and third paragraphs of the introduction, we introduce equations to formally define equivariance and invariance, and that we dedicate Figure 2 to outline this definition. As the desired property of equivariance in this work, we also argue this is made very clear throughout, and especially in the third paragraphs of page 3 where we write: “In prior work, equivariant embeddings have been shown to have numerous benefits: reduced sample complexity for training, improved generalization and transfer learning performance (Cohen & Welling, 2016b; Simeonov et al., 2021; Lenssen et al., 2018; Xiao et al., 2021). Equivariance has usually been achieved by the use of architectural modifications (Finzi et al., 2021; Cohen & Welling, 2016b), which are mostly restricted to symmetries represented as matrix groups. However, this does not cover important transformations such as photometric changes or others that cannot be represented explicitly as matrix transformations. Xiao et al. (2021) and Dangovski et al. (2022) propose more flexible approaches to allow for using more general input transformations. However, a key distinction between these works and ours is that we parameterize the transformations M in latent space and learn them, allowing for steering.”
> > >
> > > In response to “ Further, I highly suggest the authors to carefully revise the writing”. We are confused by what this means. As it stands, we believe the manuscript is largely typo-free and that following revisions thanks to reviews, the exposition is now clear and readable. If there are specific areas of the writing the reviewer would like to see revised, we welcome suggestions.

---

### Author Response · Authors · 2023-05-26
**Response to all reviewers and revised Manuscript**

We thank all reviewers for their time and valuable feedback that has helped make our paper better. We have incorporated the reviewer comments and modified our manuscript accordingly. Reviewers can find the edited sections highlighted in purple for easier perusal.

---

### Decision · Action_Editors · 2023-08-19

**Recommendation:** Reject

**Comment:**

The paper studies a downside effect of the popularly used data augmentations on some downstream tasks whose learned representations should instead be equivariant to some data augmentations. The paper proposes a representation learning method that aims to achieve such equivariance by the use of steerable representations, and the representations can be manipulated in the embedding space via learned linear maps. Experiments show some efficacy of the proposed method on its aimed task settings.

However, both reviewers Q7EF and PSUN are concerned with whether the aimed equivariance or steerability is indeed achieved by the proposed method, and argue that the experiments including OOD results reported in the paper cannot fully support the claims, and the improvements may be due to other factors. Reviewer Q7EF suggests using alternative equivariant measurement that can better analyze the learned representation. They are also concerned with the paper writing and visualizations.

In the rebuttal phase, the authors have responded to some of these concerns, including giving comparative experiments that give some quantitative evidence showing that equivarance and steerability are partially achieved. However, both reviewers are not convinced by the authors' responses, mainly due to the conflict between the used simple regularization term and 2d data augmentation, and the claimed achievement of equivalence and steerability. Given these facts, the paper cannot be accepted in its current form.

**Audience:**

Yes.

**Claims And Evidence:**

Not exactly. This is a main factor due to which two of the three reviewers cannot support acceptance of this paper. Please refer to the following comment for the details.